# Class-Disentanglement and Applications in Adversarial Detection and Defense

**Kaiwen Yang[1], Tianyi Zhou[2,3], Yonggang Zhang[1], Xinmei Tian[1], Dacheng Tao[4]**
University of Science and Technology of China[1]; University of Washington, Seattle[2]
University of Maryland, College Park[3]; JD Explore Academy[4]
{kwyang, yonggang}@mail.ustc.edu.cn, xinmei@ustc.edu.cn,
tianyizh@uw.edu, dacheng.tao@gmail.com

## Abstract

What is the minimum necessary information required by a neural net $D(\cdot)$ from an image $x$ to accurately predict its class? Extracting such information in the input space from $x$ can allocate the areas $D(\cdot)$ mainly attending to and shed novel insights to the detection and defense of adversarial attacks. In this paper, we propose "class-disentanglement" that trains a variational autoencoder $G(\cdot)$ to extract this class-dependent information as $x - G(x)$ via a trade-off between reconstructing $x$ by $G(x)$ and classifying $x$ by $D(x - G(x))$, where the former competes with the latter in decomposing $x$ so the latter retains only necessary information for classification in $x - G(x)$. We apply it to both clean images and their adversarial images and discover that the perturbations generated by adversarial attacks mainly lie in the class-dependent part $x - G(x)$. The decomposition results also provide novel interpretations to classification and attack models. Inspired by these observations, we propose to conduct adversarial detection and adversarial defense respectively on $x - G(x)$ and $G(x)$, which consistently outperform the results on the original $x$. In experiments, this simple approach substantially improves the detection and defense against different types of adversarial attacks. Code is available: https://github.com/kai-wen-yang/CD-VAE.

## 1   Introduction

Deep learning provides an end-to-end solution to challenging tasks with data of complicated structures and have achieved breakthrough across different domains in recent years. However, what essential information the neural nets extract from the input and mainly rely on in producing predictions for the targeted tasks still remains mysterious. Moreover, the trained models are known to be fragile and sensitive to adversarial perturbations on the input, but how to explain this failure is still an open problem. Given entangled raw features as input, the inference of neural nets tend to remove task-redundant information and compress task-essential information to abstract concepts. On the one hand, identifying the task-essential information in the input can allocate the areas that the neural nets mainly attend to and explain the model prediction. On the other hand, this information can also be the main target of adversarial attacks, while the removed task-redundant information may suffer less distortion under attacks and still preserve clues to correct the prediction. Hence, disentanglement of these two types of information in the input (termed "task-disentanglement") may shed critical insights to address aforementioned problems.

Disentangled representations [3] have been extensively studied in unsupervised learning when training a generative model, where each dimension of the representation controls a single generative factor of data variations and changing it does not affect other factors. Probably the most studied disentanglement model is $\beta$-variational autoencoder (VAE) [20], whose objective aims to reconstruct

the input $x$ from the latent representation $z$ via $p(x|z)$ while keeping $p(z|x)$ close to the standard Gaussian $p(z) = \mathcal{N}(\mathbf{0}, \mathbf{I})$, which encourages the disentanglement of latent factors in $z$. However, VAE usually produces unrealistic and blurry images comparing to other generative models such as GAN [15]. And encouraging the disentanglement can further weaken the reconstruction performance. In addition, the learned latent factors in $z$ are often exceedingly abstract and their correspondence to specific tasks can be obscure. Hence, it is challenging to derive task-disentanglement of input from the disentangled latent representation.

Information bottleneck (IB) [42, 43, 1] can potentially provide an information-theoretic perspective of the task-disentanglement problem. IB models the information flow from input $x$ to task output $y$ through some latent representation $z$. In IB, the ideal $z$ only preserves the minimum necessary information to predict $y$ but discards all redundant information in $x$ that cannot further contribute to predicting $y$. This leads to an optimization principle that maximizes the mutual information $I(z; y)$ and meanwhile keeps $I(x; z)$ small. From IB's perspective, for supervised learning such as classification, maximizing $I(z; y)$ corresponds to minimizing the classification error of using $z$ to predict the class $y$. In unsupervised learning models like VAE, maximizing $I(z; y)$ corresponds to minimizing the reconstruction error (or maximizing the data likelihood). In both cases, an additional constraint is adopted to limit the amount of information transmitted from $x$ to $z$, forming an "information bottleneck" that filters out information redundant to the task. In $\beta$-VAE, the constraint keeps the posterior $p(z|x)$ sufficiently close to the standard Gaussian. For classification, previous researches [45, 50] argue that the hierarchical structure of neural nets encourages more information compression as going to deeper layers and the deep learning scheme (SGD + cross entropy loss) inherently solves an IB problem. However, IB has been mainly studied for latent bottleneck representations $z$ [37, 2] but underexplored to disentangle the task-relevant information in the input space. Specifically, it is non-trivial to design an IB constraint in the input space.

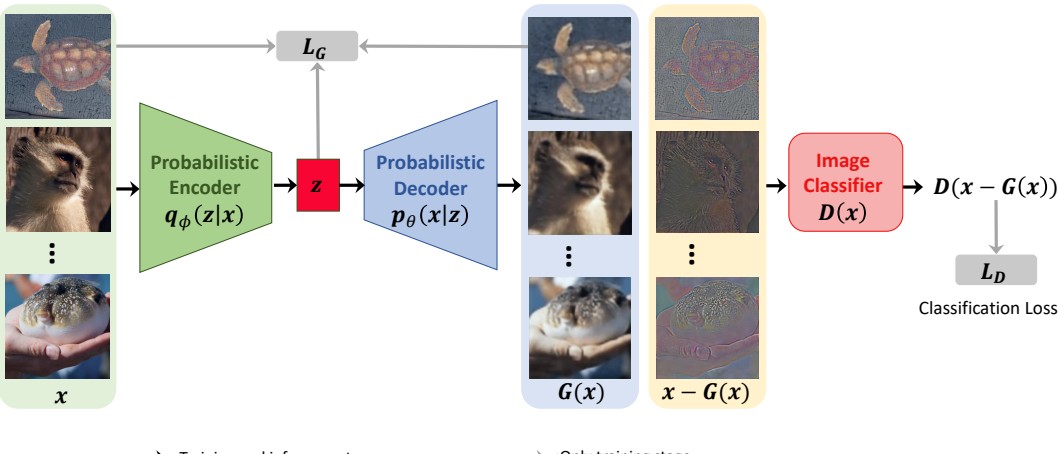

Figure 1: Architecture and Training/Inference of class-disentangled variational auto-encoder (CD-VAE).

In this paper, we focus on image classification tasks and study "class-disentanglement" as a special case of "task-disentanglement", i.e., how to decompose an input image $x$ into $x = G(x) + (x - G(x))$, where $G(x)$ contains all the information redundant or irrelevant to the class $y$, while $x - G(x)$ captures the essential information to predict the class $y$. To this end, we develop a simple architecture composed of a variational autoencoder (VAE) $G(\cdot)$ generating $G(x)$ from $x$ and a classification network $D(\cdot)$ that can be applied to both $x - G(x)$ and $x$. A class-disentanglement model is achieved by jointly training $G(\cdot)$ to reconstruct $x$ and $D(\cdot)$ to classify $x - G(x)$, where each task forms an IB constraint for the other in the objective. Therefore, $x - G(x)$ only competes the essential information for classification with $G(x)$ and $G(x)$ preserves all the class-redundant information for better reconstruction.

Unlike disentanglement of latent factors and IB defined in a latent space [5, 10, 24], class-disentanglement performs disentanglement of class-essential and class-redundant information in the more intricate but interpretable input space. Hence, it can be applied to analysing the behaviors of neural nets and intrinsic properties of complicated data, shedding novel insights that cannot be revealed by latent factors. For example, $x - G(x)$ can highlight the areas in an image $x$ that neural nets mainly attend to and help explain why correct or incorrect predictions are made. In this paper,

we mainly explore its applications on analyzing adversarial examples. In particular, we find that the adversarial perturbations generated by mainstream attack algorithms primarily lie in $x - G(x)$, while $G(x)$ still preserves uncontaminated information useful for classification. Therefore, we propose to detect adversarial attacks on $x - G(x)$ and defend the attacks by make predictions based on $G(x)$. In extensive experiments on both CIFAR and ImageNet dataset, these simple strategies significantly improve previous approaches directly applied on $x$ and additionally provide novel interpretations to the results. We also present an empirical study of the trade-off between reconstruction and classification when changing their weights in the objective.

## 2  Related Work

**Disentangled Representation** Different variants of VAE [5, 10, 20, 24] have been developed in recent years for learning disentangled latent representations. They all rely on unsupervised learning to automatically discover explanatory factors of variation. However, as suggested by the theoretical analysis in [33], unsupervised disentanglement is fundamentally impossible without proper inductive biases on models and data. Hence, (implicit) supervision is necessary and might provide crucial guidance to disentanglement, as shown in class-disentanglement of this paper as well. Inspired by image translation, disentangling the content and style of an image is studied in [32, 4, 14, 22]. These methods leverage the supervised information, i.e., the domains or groups that every sample belongs to. Unlike previous works, class-disentanglement in this paper combines unsupervised learning and supervised learning in one objective to train a disentanglement model. These two tasks compete with each other and form IB constraints to each other, which leads to efficient disentanglement of class-essential and class-redundant information. Class-disentanglement of latent representations has been studied in previous works [47, 18], which however fundamentally differ from our input-space disentanglement on optimization formulations, model architectures, and/or applications. More detailed comparison can be found in Sec. I of the Appendix. The input-space class-disentanglement can be explained as a non-trivial extension of Robust PCA [6, 52], which decomposes a data matrix into a low-rank part and a sparse part.

**Adversarial Detection** aims to distinguish adversarial examples from natural examples. Given a neural net trained on a clean dataset for the original task, many existing methods [36, 17, 9] train a binary classification network on top of some intermediate-layer of the given network as the adversarial detector. However, as pointed out by [7], these methods may fail when the detector model is leaked. Another strategy is motivated by the observation that adversarial examples have very different distribution from natural examples on intermediate-layer features. So a detector can be built upon some statistics of the distribution, e.g., [13] estimates the kernel density of an example as its proximity to the natural examples' manifold; [30] models the natural examples' distribution by a multivariate Gaussian and the Mahalanobis distance of an example to this Gaussian is used for adversarial detection; [34] introduces local intrinsic dimensionality to describe the distribution of adversarial examples. Most of these methods directly apply a detection on the input example $x$. Instead, we propose to conduct the detection on the $x - G(x)$ capturing the essential information of the class. This simple method consistently improves the detection accuracy and is complementary to existing approaches.

**Adversarial Defense** is usually achieved by adversarial training [35] and its variants [49] that augment the training set with adversarial examples. Another type of adversarial defense trains a GAN [41] or an auto-encoder [31]to remove adversarial perturbations (if any) from a sample before sending it to the classifier. For instance, [41] proposes an adversarial perturbation erasing GAN (APE-GAN) that can modify an adversarial example close to a natural example; [31] proposed a high-level representation guided denoiser (HGD) to remove adversarial noise. In this paper, we show that classification on the class-redundant part $G(x)$ is robust to adversarial attacks. So our method falls into this category. However, it differs from APE-GAN and HGD as we do not utilize adversarial examples during training.

## 3  Class-Disentanglement

In this section, we first introduce a simple VAE-classifier architecture and an associated optimization objective to train it for class-disentanglement. We then conduct an empirical study that applies the class-disentanglement model to both clean images and their adversarial examples. The results show that the adversarial perturbations mainly reside in the class-essential part $x - G(x)$. These

observations motivate us to conduct adversarial detection and defense on $x - G(x)$ and $G(x)$, respectively, which are elaborated at the end of this section.

## 3.1 Class-Disentangled Variational Auto-Encoder

We propose a model architecture shown in Fig. 1 to decompose an input image $x$ into a part retaining only essential information for classification (i.e., class-essential) and a part covering all the rest redundant information (i.e., class-redundant). Adding the two parts together exactly recovers $x$. In the proposed "class-disentangled variational auto-encoder (CD-VAE)", we employ a variataional auto-encoder (VAE) $G(\cdot)$ to capture the class-redundant information $G(x)$ and then send the complementary part $R(x) \triangleq x - G(x)$ to a classification network $D(\cdot)$. They are trained jointly: VAE $G(\cdot)$ aims for reconstructing $x$ while classifier $D(\cdot)$ aims for predicting the class of $x$ from $x - G(x)$. Therefore, the reconstruction and classification compete with each other on the input pixels of $x$ and thus discourage sending redundant information to each other. From the information bottleneck perspective, each of them places an IB constraint for the information flowing to the other.

In CD-VAE, VAE $G(\cdot)$ is composed of (1) an encoder parameterized by $\phi$ and producing the posterior $q_\phi(z|x)$ of latent factors $z$; and (2) an decoder parameterized by $\theta$ and producing the data likelihood $p_\theta(x|z)$. Classification network $D(\cdot)$ predicts the class probabilities for $x$ based on $x - G(x)$ and it has parameters $\omega$. We jointly train $G(\cdot)$ and $D(\cdot)$ by optimizing the following objective, which combines the objective $L_G(\phi, \theta)$ of $\beta$-VAE and the cross-entropy loss $L_D(\omega)$ for classification, i.e.,

$$\min_{\phi,\theta,\omega} \mathbb{E}_{(x,y)\sim p_{data}(x,y)} \left[ L_G(\phi, \theta) + \gamma L_D(\omega) \right] \tag{1}$$

$$L_G(\phi, \theta) = -\mathbb{E}_{q_\phi(z|x)} \log p_\theta(x|z) + \beta D_{KL}(q_\phi(z|x) \| p(z)), \tag{2}$$

$$L_D(\omega) = -\log D(x - G(x); \omega)[y], \tag{3}$$

where $p(z) = \mathcal{N}(\mathbf{0}, \mathbf{I})$ is the prior of $z$ that encourages its disentanglement, and $D_{KL}(\cdot \| \cdot)$ measures the Kullback–Leibler divergence of posterior $q_\phi(z|x)$ from prior $p(z)$. The first term in $L_G(\phi, \theta)$ is the marginal likelihood of data in the VAE-generation process. It aims for reconstructing $x$ under the IB constraint of the KL-divergence. In $L_D(\omega)$, $D(x - G(x); \omega)[y]$ is the predicted probability for the ground-truth class $y$. There are two hyperparameters in the CD-VAE objective: $\beta$ controls the strength of representation-disentanglement inside the $\beta$-VAE, while $\gamma$ controls the trade-off between reconstruction task (by $G(x)$) and classification task (by $D(x - G(x))$).

Not every pixel in an image $x$ is related to the class label $y$ and many previous works [51, 40] imply that a neural net classifier makes predictions by merely attending to sparse regions of the image. However, the rest information outside the scope of classifiers, e.g., illumination, background colors, patterns shared with other classes, objects of other classes, etc., are all critical to the reconstruction of $x$. Hence, the trade-off between VAE $G(\cdot)$ and classifier $D(\cdot)$ in CD-VAE is efficient for class-disentanglement: while the classifier $D(\cdot)$ quickly picks the class-specific regions in $x$, the class-redundant information more general across classes can be easily captured by $G(\cdot)$'s bottleneck factors $z$ with higher priority. The optimization of Eq. (1) can converge fast in a few epochs, as shown in Sec. F of the Appendix.

## 3.2 Class-Disentanglement on Natural Images vs. Adversarial Images: An Empirical Study

CD-VAE provides novel perspectives to understand (1) how a neural net classifier predicts the class of an image and (2) how an adversarial example attacks the classifier. In particular, $x - G(x)$ reveals the regions in an image that a classifier mainly attend to, and our empirical study shows that they are also the regions that adversarial examples mainly attack. Since adversarial perturbations are optimized and generated in the input space, class-disentanglement in the input space is more adapted for their analysis than representation-disentanglement.

Given a pre-trained classifier $\hat{D}(\cdot)$ [1] and a clean image $x$ with label $y$, an adversarial image $x' = x + \delta$ is usually generated by optimizing $\delta$ to maximize the classification loss within an $\epsilon$-ball around $x$, i.e.,

$$\max_{\|\delta\|_p \leq \epsilon} -\log \hat{D}(x + \delta)[y], \tag{4}$$

---

[1]Here $\hat{D}(\cdot)$ differs from $D(\cdot)$ in CD-VAE and is a classifier pre-trained on clean data $x$.

**Original label:** Vizsla, Macaw, Airedale, King crab, English springer, Gibbon, Weimaraner, Saluki

**Predicted label after attack:** Afghan hound, Lorikeet, Otterhound, Rock crab, Cocker spaniel, Patas, Chesapeake bay retriever, Basenji

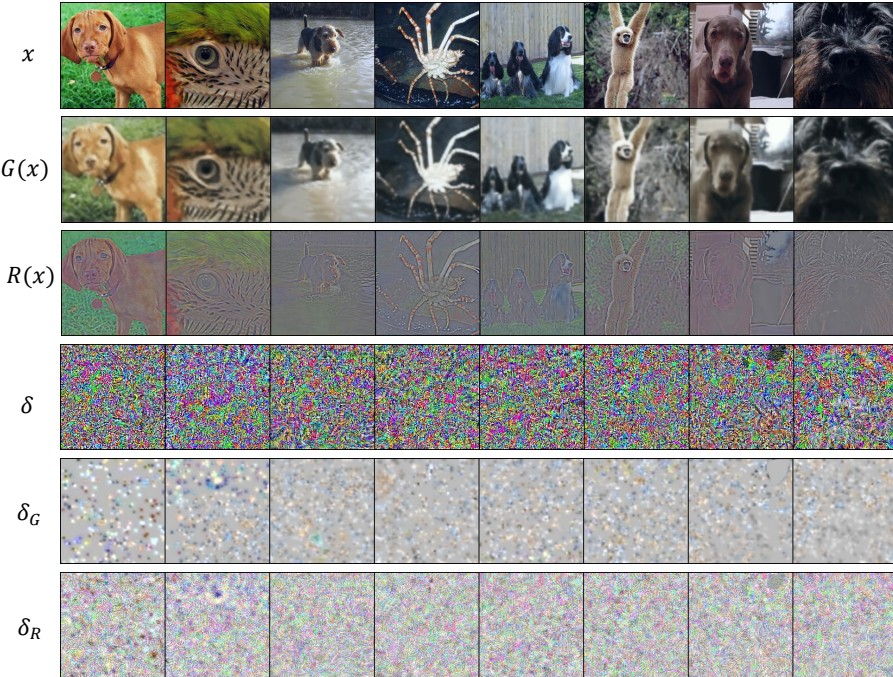

Figure 2: Class-Disentanglement results by CD-VAE on restricted ImageNet.

where the popular choices for $p$ include $p = \infty, 1, 2$. This is called "non-targeted" attack since it does not specify a target class that the prediction should change to after applying the perturbation $\delta$. A widely used algorithm to tackle the above optimization is projected gradient descent (PGD), which iterates between gradient sign ascent of step size $\alpha$ and projection $\mathcal{P}_{\|x'-x\|_p \leq \epsilon}(\cdot)$ to the $\epsilon$-ball around $x$ for multiple steps, i.e.,

$$x' \leftarrow \mathcal{P}_{\|x'-x\|_p \leq \epsilon} \left( x' + \alpha \operatorname{sign} \left( \nabla_{x'} - \log \hat{D}(x')[y] \right) \right). \tag{5}$$

Although there exist other types of adversarial attacks that can also be analyzed by class-disentanglement, we mainly use PGD attack here and will include more attacks in later experiments.

In our empirical study, we compare the class-disentanglement results of a clean images $x$ and an adversarial image $x'$ generated from it, i.e., $G(x)$, $R(x) = x - G(x)$, $G(x')$, $R(x') = x' - G(x')$. We are particularly interested in (1) which parts of an image suggested by $R(x)$ and $R(x')$ that a neural net classifier mainly attends to; and (2) the difference between $x$ and $x'$ on their original pixels, class-essential parts, class-redundant parts and hidden space. Specifically, we investigate

$$\delta \triangleq x - x', \ \delta_G \triangleq G(x) - G(x'), \ \delta_R \triangleq R(x) - R(x'), \ \delta_z \triangleq z - z'. \tag{6}$$

For each dataset from CIFAR-10 and ImageNet, we train a CD-VAE model on the (clean) training set. We then apply the CD-VAE model to both the (clean) test/validation set images and the corresponding adversarial images generated by PGD attack ($\ell_\infty$-ball constraint)

|  | Consine Similarity | FID |
|---|---|---|
| $G(x)$ vs. $G(x')$ | $0.9997 \pm 0.0003$ | 0.90 |
| $R(x)$ vs. $R(x')$ | $0.9460 \pm 0.0260$ | 22.31 |

Table 1: Cosine similarity and FID.

[35] towards a pre-trained classifier (WideResNet-28-10 [48] for CIFAR-10 and ResNet-50 [19] for ImageNet). Fig. 2 presents examples of the ImageNet experiment. Similar results for CIFAR-10 can be found in Appendix.

It shows that the class-essential part $R(x) = x - G(x)$ only highlight sparse regions of each image but can capture critical features of the class that are sufficient for the classification task, e.g., ears and nose of dog, eyes of parrot, etc. Hence, $R(x)$ can provide a natural interpretation

to the prediction made by the neural net classifier, which has complicated structure and is usually treated as a black box. Comparison among $\delta$, $\delta_R$ and $\delta_G$ shows that the adversarial perturbation $\delta$ mainly lies in $\delta_R$ and cause slight change $\delta_G$ on $G(x)$. This indicates that the attacks to $x$ mainly reside in $R(x')$ but do not heavily distort $G(x)$. This observation directly leads to two applications. First, we may better detect adversarial examples in the space of $R(x)$. In Fig. 2, the sparse regions captured by $R(x')$ largely narrow the search range for the attacked regions and thus might make the detection easier. Secondly, $G(x)$ in Fig. 2 still contains useful information for class prediction—it is redundant and less discriminative than $R(x)$ but is almost undistorted in $G(x')$, so classification on $G(x')$ might be more robust to adversarial attacks and provide a simple defense strategy.

Our quantitative analysis to $\delta$, $\delta_R$ and $\delta_G$ also support the observations above. In Table 2, we report several statistics of them, e.g., the mean and standard deviation for their $\ell_p$ norms with $p = \{1, 2, \infty\}$. To further study the similarity of $R(x)$ and $R(x')$ as images, we also report the statistics for their cosine similarity and Fréchet Inception Distance (FID) in Table 1. $R(x')$ and $R(x)$ are distant from each other while $G(x')$ is close to $G(x)$ in terms of all those distance

|  | $\ell_1 \times 10^{-3}$ | $\ell_2$ | $\ell_\infty$ |
|---|---|---|---|
| $x$ | $142.83 \pm 37.02$ | $427.99 \pm 92.59$ | $2.49 \pm 0.20$ |
| $G(x)$ | $133.68 \pm 37.07$ | $397.77 \pm 92.14$ | $2.08 \pm 0.11$ |
| $R(x)$ | $36.76 \pm 11.75$ | $127.95 \pm 37.27$ | $2.45 \pm 0.46$ |
| $x'$ | $142.74 \pm 36.57$ | $427.91 \pm 91.19$ | $2.53 \pm 0.18$ |
| $G(x')$ | $133.52 \pm 36.96$ | $397.26 \pm 91.85$ | $2.08 \pm 0.11$ |
| $R(x')$ | $38.13 \pm 11.06$ | $131.20 \pm 35.37$ | $2.46 \pm 0.45$ |
| $\delta$ | $13.65 \pm 0.88$ | $39.09 \pm 1.40$ | $0.14 \pm 0.00$ |
| $\delta_G$ | $4.15 \pm 0.65$ | $16.56 \pm 2.47$ | $0.39 \pm 0.17$ |
| $\delta_R$ | $13.78 \pm 0.92$ | $40.97 \pm 1.87$ | $0.48 \pm 0.16$ |

Table 2: $\ell_p$ norm/distance of different parts in CD-VAE (ImageNet).

metrics. This again supports the adversarial detection on $R(x)$ and adversarial defense on $G(x)$. In addition, we calculate the (Pearson) correlation between $z$ and $z'$ on each dimension and report the its histogram and $p$-value in Fig. 3. It shows a strong correlation between $z$ and $z'$, which implies that $G(x)$ is robust to the adversarial perturbations. An extensive quantitative analysis of $\delta$, $\delta_G$ and $\delta_R$ with respect to larger $\epsilon$ is provided in Sec. E of the Appendix.

| dataset | Training \ Test | $x$ | $R(x)$ | $G(x)$ |
|---|---|---|---|---|
| CIFAR-10 | $x$ | 96.01(99.84) | 92.68(99.65) | 18.86(67.93) |
| | $R(x)$ | 95.81(99.81) | 96.20(99.82) | 18.12(66.30) |
| | $G(x)$ | 51.84(86.52) | 25.67(68.98) | 75.25(97.39) |
| Restricted ImageNet | $x$ | 82.53(96.88) | 11.17(25.61) | 65.26(88.94) |
| | $R(x)$ | 63.33(82.14) | 82.14(96.47) | 5.36(12.01) |
| | $G(x)$ | 44.46(69.13) | 1.71(5.02) | 74.08(93.37) |

Table 3: Training on one part of CD-VAE (using large $\gamma = 2$) and test on another part: Top-1(Top-5)

In Table 3, in order to further understand how the class-information is distributed into $R(x)$ and $G(x)$ and why the adversarial attacks towards $x$ mainly affect $R(x)$ instead of $G(x)$, we train three classifiers respectively on $x$, $G(x)$ and $R(x)$ of training images and evaluate their accuracy when applied to $x$, $G(x)$ and $R(x)$ of test images. The results are reported in Table 3, from which we observe: (1) the classifier trained on $R(x)$ is highly effective on $x$ and vice versa (with only one exception) so they share class-information important to both classifiers; (2) the classifier trained on $G(x)$ has some classification accuracy, indicating that $G(x)$ still preserves useful information about class even with large

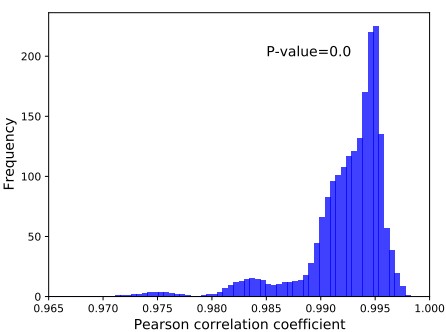

Figure 3: Pearson correlation of $z$ and $z'$.

$\gamma$. But the classifier trained on $x$ performs poorly on $G(x)$ so it ignores and does not rely on the class-information in $G(x)$; (3) the classifier trained on $R(x)$ achieves poor accuracy on $G(x)$ and vice versa, indicating that they use different information for classification. Therefore, adversarial attacks to the classifier trained on $x$ will mainly distort the class-information shared by and important to both $x$ and $R(x)$ but slightly affect the class-information in $G(x)$.

### 3.3 Applying Class-Disentanglement to Adversarial Detection

Inspired by the empirical study, we propose to apply existing adversarial detection methods to $R(x) = x - G(x)$ instead of $x$. For example, given an image $x$, [30] builds a kernel density estimator $K(\cdot, \hat{D}_\ell(X))$ for the intermediate-layer embedding (e.g., the $\ell^{th}$ layer output $\hat{D}_\ell(\cdot)$ of a classifier $\hat{D}(\cdot)$) of the (clean) training set $X$ and use it to compute the kernel density of a given sample $x$, i.e., $K(x, \hat{D}_\ell(X))$. Intuitively, adversarial examples will have smaller density than the natural samples. Hence, previous works detect them via thresholding to the density value. Instead, we propose to apply the same method to $R(x)$ computed by CD-VAE, i.e., we detect sample $x$ as an adversarial example if its kernel density $K(R(x), D_\ell(R(X)))$ is below a threshold[2]. Note this simple strategy is complementary to existing techniques for adversarial detection methods and thus can be generally applied to improving them.

### 3.4 Applying Class-Disentanglement to Adversarial Defense

**Defense against Grey-box attack**    As revealed in Table 2-3, $G(x)$ preserves information useful for classification but redundant and orthogonal to the class information in $R(x)$, which can be mainly distorted by adversarial attacks against the classifier pre-trained on $x$. This leads to a simple defense strategy: given an adversarial example $x'$ generated by adversarial attacks to a classifier trained on $x$, we can use $G(x')$ generated by CD-VAE for robust classification since the class information in $G(x')$ is less distorted by attacks. Note this strategy holds for grey-box attacks that have access to the pre-trained classifier but not the CD-VAE model.

**Defense against White-box attack**    We further develop an adversarial training method of CD-VAE as a defense against white-box attacks [8] that can access both the classifier and the whole CD-VAE model. In Section 3.4, we propose to classify adversarial images by a classifier trained on $G(x')$. Hence, white-box attacks generate an adversarial image $x' = x + \delta$ by optimizing the perturbation $\delta$ towards maximizing the classification loss of $G(x')$ within an $\ell_p$ $\epsilon$-ball around $x$, i.e.,

$$\max_{\|\delta\|_p \leq \epsilon} -\log D_G(G(x + \delta))[y], \tag{7}$$

where $D_G$ is the classifier trained on $G(x')$ and $y$ is the true label. Under white-box attacks, the perturbation $\delta$ tends to mainly distort $G(x)$ to deceive the classifier $D_G(\cdot)$, so $\delta_G$ could dominate $\delta$ and $R(\cdot)$ is no longer the main focus oe attacks. Unlike existing adversarial training methods, which train the whole model (i.e., CD-VAE and $D_G$ in our case) to correctly classify $x'$, class-disentanglement model in CD-VAE enables a novel and more effective adversarial training strategy. In particular, for samples whose predictions by $D_G$ are heavily distorted, i.e., the true-class probability is similar or lower than the probability of a wrong class: $D_G(G(x'))[y] - \max_{y' \neq y} D_G(G(x'))[y'] \leq c$ with margin $c$, we train $D_G(G(x'))$ to maximize its predicted probability on the true class $y$ and meanwhile train $D(R(x'))$ (i.e., the classifier in CD-VAE) to maximize its predicted probability on the most probable wrong-class $\text{argmax}_{y' \neq y} D_G(G(x'))[y']$. Thereby, besides encouraging $D_G(G(x'))$ to correctly predict $y$, it enforces the distorted class information in $x'$ or $G(x')$ to move to $R(x')$ instead of $G(x')$. Hence, $G(x')$ can preserve the class-related information robust to adversarial attacks. Formally, the original CD-VAE objective is modified with a new $L_D$, i.e.,

$$\min_{\phi,\theta,\omega,\omega_G} \mathbb{E}_{\{(x',y): D_G(G(x'))[y] - \max_{y' \neq y} D_G(G(x'))[y'] \leq c\}} [L_G(\phi,\theta) + \gamma L_D(\omega,\omega_G)] \tag{8}$$

$$L_G(\phi,\theta) = -\mathbb{E}_{q_\phi(z|x')} \log p_\theta(x'|z) + \beta D_{KL}(q_\phi(z|x')\|p(z)), \tag{9}$$

$$L_D(\omega,\omega_G) = -\log D_G(G(x');\omega_G)[y] - \log D(R(x');\omega)[\text{argmax}_{y' \neq y} D_G(G(x'))[y']], \tag{10}$$

where $x'$ is generated from $x$ by attacks in Eq. (7), $L_G(\phi,\theta)$ is the same as Eq. (2) in the original objective , and $L_D(\omega,\omega_G)$ is composed of the two losses associated with the above strategy. By optimizing this objective, CD-VAE can disentangle the mainly distorted class-information and the class-information robust to the white-box attack within $x'$, and assign them to $R(x')$ and $G(x')$ respectively. In practice, we start from a CD-VAE pre-trained on clean images and then mix the original CD-VAE training (i.e., Eq. (1)-(3)) on clean images with the above adversarial training on strong adversarial examples $x'$ within margin $c$.

---

[2]Here $D(\cdot)$ is the classifier in CD-VAE trained on $R(x)$ and adversarial example $x'$ attacks $D(\cdot)$ in CD-VAE.

# 4 Experiments

In this section, we evaluate the classification and reconstruction in CD-VAE under different $\gamma$ and apply CD-VAE to adversarial detection and defense of a variety of attacks, as suggested by the empirical study above. We further compare CD-VAE with popular approaches developed for the two tasks and show consistent improvement over them brought by our strategies proposed in Sec. 3.3-3.4. In particular, all the experiments are conducted on two datasets, CIFAR-10 [26] and restricted ImageNet [44]. CIFAR-10 consists of 60,000 color images belonging to 10 classes, with 50,000 training images and 10000 test images. Restricted ImageNet [44] is an ImageNet [11] subset with 203 classes including 257,748 training images and 10,150 test images.

**CD-VAE Setting** For CIFAR-10, we use a VAE $G(\cdot)$ with a few convolutional layers [25] and WideResNet-28-10 [48] as the image classifier $D(\cdot)$. For restricted ImageNet, due to the high resolution ($299 \times 299$), we use VQ-VAE [38] as $G(\cdot)$ and ResNet-50 [19] as $D(\cdot)$. We set $\beta = 0.2$. During training, we use AdamW [12] with a weight decay of 1e-6 as the optimizer to minimize Eq. (1) in an end-to-end manner for 300(60) epochs on CIFAR-10(restricted ImageNet).

## 4.1 Trade-off between Reconstruction and Classification in CD-VAE (via $\gamma$)

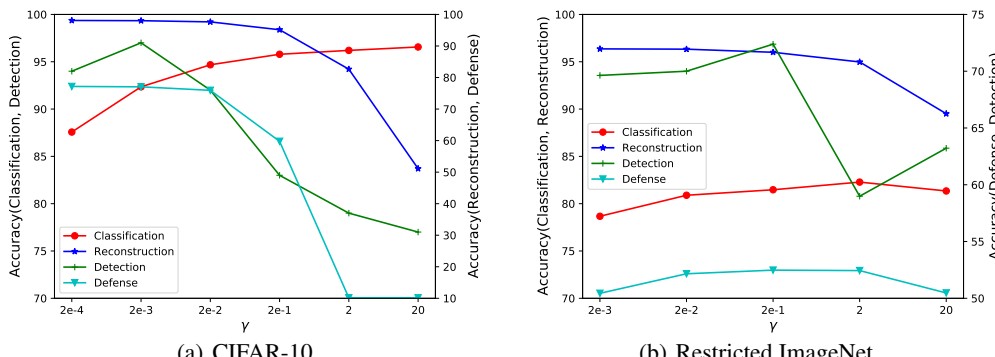

(a) CIFAR-10            (b) Restricted ImageNet

Figure 4: Trade-off between reconstruction accuracy and classification accuracy by changing $\gamma$.

In the CD-VAE objective Eq. (1), $\gamma$ controls the trade-off between classification on $R(x) = x - G(x)$ and reconstruction by $G(x)$. When applied to adversarial detection and defense, it may lead to more complicated trade-off behaviors. In Fig. 4, we evaluate classification accuracy of $D(R(x))$, reconstruction accuracy on $G(x)$ (i.e., $1 - \|x - G(x)\|_2/\|x\|_2$), adversarial detection accuracy, and the defense accuracy on adversarial examples, when using different $\gamma$ to train CD-VAE.

In both plots, classification (reconstruction) accuracy quickly grows when increasing (decreasing) $\gamma$ from small (large) values and then saturates after sufficiently large (small) $\gamma$, indicating that moving more information to $R(x)$ ($G(x)$) cannot improve the classification (reconstruction). Therefore, for both datasets, a sweet spot is achieved at $\gamma = 2$ for the classification-reconstruction trade-off. Changing $\gamma$ also leads to the changes of visual patterns in $R(x)$ and $G(x)$, as shown in Fig. 5 in Sec. A of the Appendix, which reveals the priority of different class information. Fig. 4 also reports how the adversarial detection and defense accuracy changes with $\gamma$ (against PGD-attack to a pre-trained classifier on the training set). They does not change monotonically with $\gamma$ since increasing (decreasing) $\gamma$ may move both perturbation-specific and true class-specific information into $R(x)$ ($G(x)$): while the former can improve the detection, the latter introduces noises.

## 4.2 Adversarial Detection

We apply our strategy in Sec. 3.3 to several state-of-the-art adversarial detection approaches, i.e., kernel density (KD) [13], local intrinsic dimensionality (LID) [34] and mahalanbobis distance (MD) [30], and compare it with their original versions on the tasks of detecting adversarial examples generated by several state-of-the-art adversarial attacks, i.e., fast gradient sign method (FGSM)[16], basic iterative method (BIM)[27], Carlini& Wagner's method (C&W)[8] and projected gradient descent method (PGD)[35], where $\ell_2$-ball and $\ell_\infty$-ball $\epsilon$-constraints (with $\epsilon = 1$ and $8/255$) are both considered, and iterative attacks run for 5 steps using step size $\frac{\epsilon}{\sqrt{2}}$. We use WideResNet-28-10 (ResNet-50) as the DNN feature extractor for CIFAR-10 (restricted ImageNet), which is $D(\cdot)$ from

| Method | FGSM | | BIM | | C&W | | PGD-$l_\infty$ | | PGD-$l_2$ | |
|---|---|---|---|---|---|---|---|---|---|---|
| | TNR | AUC | TNR | AUC | TNR | AUC | TNR | AUC | TNR | AUC |
| KD | 42.38 | 85.74 | 74.54 | 94.82 | 73.33 | 94.75 | 73.12 | 94.59 | 70.62 | 93.62 |
| KD ($R(x)$) | 57.10 | 89.69 | 96.79 | 99.27 | 94.67 | 98.73 | 96.56 | 99.30 | 97.04 | 99.32 |
| LID | 69.05 | 93.60 | 77.73 | 95.20 | 74.98 | 94.32 | 71.52 | 93.19 | 72.57 | 93.46 |
| LID ($R(x)$) | 92.60 | 98.59 | 86.42 | 97.29 | 76.42 | 95.10 | 87.54 | 97.57 | 87.63 | 97.38 |
| MD | 94.91 | 98.69 | 88.33 | 97.66 | 86.30 | 97.36 | 77.23 | 95.38 | 76.70 | 95.33 |
| MD ($R(x)$) | 99.68 | 99.36 | 98.92 | 99.74 | 98.94 | 99.68 | 99.13 | 99.79 | 99.13 | 99.77 |

Table 4: TNR and AUC (%) of adversarial detection on $x$ vs. $R(x)$ (ours) against 5 attacks (CIFAR-10)

| Method | FGSM | | BIM | | C&W | | PGD-$l_\infty$ | | PGD-$l_2$ | |
|---|---|---|---|---|---|---|---|---|---|---|
| | TNR | AUC | TNR | AUC | TNR | AUC | TNR | AUC | TNR | AUC |
| KD | 8.76 | 70.25 | 99.26 | 99.82 | 99.22 | 99.77 | 99.25 | 99.80 | 44.02 | 82.81 |
| KD ($R(x)$) | 99.40 | 99.63 | 100.0 | 100.0 | 100.0 | 100.0 | 100.0 | 100.0 | 92.85 | 98.40 |
| LID | 44.28 | 85.91 | 97.92 | 99.57 | 97.58 | 99.49 | 97.80 | 99.51 | 53.78 | 87.37 |
| LID ($R(x)$) | 99.91 | 99.93 | 99.46 | 99.81 | 98.24 | 99.40 | 99.43 | 99.73 | 61.32 | 89.18 |
| MD | 34.69 | 84.96 | 99.33 | 99.78 | 98.72 | 99.61 | 99.18 | 99.70 | 44.16 | 82.61 |
| MD ($R(x)$) | 100.0 | 99.99 | 100.0 | 99.99 | 100.0 | 99.98 | 99.99 | 99.95 | 95.42 | 98.86 |

Table 5: TNR and AUC (%) of adversarial detection on $x$ vs. $R(x)$ (ours) against 5 attacks (restricted ImageNet)

CD-VAE for our strategy but is trained on the training set for all other baselines. We use true negative rate (TNR) at 95% true positive rate and area under the receiver operating characteristic curve (AUC) to evaluate the detection performance. The results on CIFAR-10 and restricted ImageNet are reported in Table 4-5. By applying the existing detection methods on $R(x)$ instead of $x$, our strategy consistently and significantly improves all these methods' detection accuracy. For example, FGSM is the most challenging attacks to detect but our method can still improve TNR of KD from 8.76% to 99.40% on restricted ImageNet. White-box detection's performance on other datasets and its generalization to other unseen attacks are reported in Sec. B and Sec. C of the Appendix.

## 4.3 Adversarial Defense against Grey-Box Attacks

We evaluate our adversarial defense strategy in Sec. 3.4 against several state-of-the-art attacks, i.e., gradient-based attack (BIM[27], PGD[35] and C&W[8]) and spatial transform-based attacks (StAdv[46]), with $\epsilon = 1$, $8/255$, $0.05$ for $\ell_2$, $\ell_\infty$, and StAdv, respectively. We set the iterations of attacks to 100 and the attacks are applied against pre-trained WideResNet-28-10 (Resnet50) on CIFAR-10 (restricted ImageNet). According to the trade-off in Fig. 4, we choose $\gamma = 2e - 3$ for CIFAR-10 and $\gamma = 2e - 1$ for restricted ImageNet. We compare our method with adversarial training based methods and preprocessing based methods (HGD[31], APE-GAN[41]).

The defense accuracy over adversarial examples are reported in Table 6. Our strategy outperforms most baselines by a large margin when defending all types of attacks, while other methods only excel in defending certain attacks, e.g., adversarial training (AT) methods excel on the attacks used in AT but generalize poorly to other attacks. HGD is the most competitive one among pre-processing based methods but our method consistently outperforms it. Comparing to other baselines relying on adversarial examples for training, CD-VAE trained only on natural examples already achieves higher defense accuracy. Note the degeneration of clean data accuracy in Table 6 is due to the widely observed trade-off between it and the robustness. In CD-VAE, as evaluated in Sec 4.1, such trade-off can be controlled by tuning $\gamma$.

## 4.4 Adversarial Defense against White-Box Attacks

We evaluate our adversarial defense strategy towards white-box defense under several state-of-the-art white-box attacks on CIFAR-10, the experiment details of which are given in Sec. D in the Appendix. We report the defense accuracy in Table. 7. Our method achieves the highest "Unseen Attacks (mean)", which is the defense accuracy averaged over all the attacks that are not used for adversarial training of the defense model. It also achieves the highest defense accuracy against two out of the five attacks,

| Dataset | Defense | Attack | | | | | |
|---------|---------|-------|--------|--------|--------|--------|-------|
| | | Clean | PGD-$\ell_\infty$ | PGD-$\ell_2$ | C&W-$\ell_\infty$ | C&W-$\ell_2$ | StAdv |
| | Normal | 96.01 | 0.0 | 0.0 | 0.0 | 0.0 | 0.0 |
| | AT PGD-$\ell_\infty$ | 86.8 | 51.7 | 24.3 | 52.0 | 26.0 | 4.8 |
| | TRADES $\ell_\infty$ | 84.9 | 55.1 | 28.0 | 53.8 | 28.3 | 9.2 |
| | AT PGD-$\ell_2$ | 85.0 | 41.9 | 50.1 | 43.4 | 50.6 | 7.8 |
| CIFAR10 | AT StAdv | 86.2 | 0.1 | 0.3 | 0.2 | 0.5 | 53.9 |
| | HGD | 80.75 | 75.93 | 75.44 | 75.84 | 77.15 | 23.04 |
| | APE-GAN | 90.93 | 59.28 | 65.17 | 59.23 | 65.30 | 7.28 |
| | Ours | 86.81 | 77.05 | 78.02 | 77.04 | 78.29 | 19.41 |
| | Normal | 82.53 | 0.0 | 0.0 | 0.0 | 0.0 | 0.0 |
| ImageNet | AT PGD-$\ell_2$ | 69.89 | 10.93 | 60.95 | 9.49 | 60.07 | 0.31 |
| | Ours | 65.26 | 52.48 | 63.12 | 52.95 | 64.98 | 4.75 |

Table 6: Defense accuracy (%) of our strategy and baselines against various attacks. AT-adversarial training.

| Defense | Clean | Unseen Attacks (mean) | Attack | | | | |
|---------|-------|------------------------|-------------|---------|--------|---------|-------|
| | | | $\ell_\infty$ | $\ell_2$ | JPEG | ReColor | StAdv |
| Normal | 96.0 | 0.1 | 0.0 | 0.0 | 0.0 | 0.4 | 0.0 |
| AT PGD-$\ell_\infty$ | 86.8 | 27.2 | 49.0 | 19.2 | 30.2 | 54.5 | 4.8 |
| TRADES $\ell_\infty$ | 84.9 | 31.0 | 52.5 | 23.3 | - | 60.6 | 9.2 |
| AT PGD-$\ell_2$ | 85.0 | 40.3 | 39.5 | 47.8 | 60.3 | 53.5 | 7.8 |
| AT ReColorAdv | 93.4 | 7.9 | 8.5 | 3.9 | 19.2 | 65.0 | 0.0 |
| AT StAdv | 86.2 | 1.8 | 0.1 | 0.2 | 1.9 | 5.1 | 53.9 |
| HGD | 80.8 | 0.1 | 0.0 | 0.0 | 0.0 | 0.4 | 0.0 |
| APE-GAN | 90.9 | 0.2 | 0.0 | 0.0 | 0.0 | 1.1 | 0.0 |
| SAT-$\ell_\infty$(NeurIPS 2020) | 85.6 | 32.4 | 53.1 | 23.4 | 35.9 | 66.5 | 3.9 |
| OIA-$\ell_\infty$(ICML 2020) | 83.4 | 33.6 | 51.6 | 24.2 | 35.4 | 67.8 | 7.0 |
| PAT-$\ell_\infty$(ICLR 2021) | 82.4 | 50.3 | 30.2 | 34.9 | 48.7 | 71.0 | 46.4 |
| Ours-$\ell_\infty$ | 81.2 | 51.4 | 40.5 | 43.1 | 62.1 | 73.1 | 27.4 |
| Ours-$\ell_2$ | 81.0 | 50.4 | 39.4 | 42.4 | 61.6 | 72.2 | 28.4 |

Table 7: Defense accuracy (%) of our strategy and baselines against white-box attacks on CIFAR10. AT-adversarial training. "Unseen Attacks (mean)" reports the defense accuracy averaged over all the attacks that are not used for adversarial training of the defense model. Ours-$\ell_\infty$ and Ours-$\ell_2$ is trained using adversarial examples generated by C&W attacks [8] within $\ell_\infty$-ball of $\ell_2$-ball respectively.

i.e., JPEG and ReColor. For other three attacks, our model achieves comparable accuracy as the best defense model against each attack. The two preprocessing-based method, i.e., HGD and APE-GAN, both fail under all white-box attacks and achieve nearly 0% accuracy. The conventional adversarial training (AT) methods perform well in defending the model from the specific attack it is trained for but generalize poorly against unseen attacks. In contrast, our model trained merely on adversarial examples generated by C&W-$\ell_\infty$ attacks not only generalizes well towards $\ell_\infty$ attack but also show much better robustness towards other types of unseen attacks. In addition, we compare our method with three latest state-of-the-art methods, i.e., SAT[21], OIA[39] and PAT[29]. Even when compared to their best model checkpoints, our method still outperforms them by a large margin on the defense accuracy against unseen attacks, because CD-VAE can automatically move the class information primarily distorted by adversarial attacks from $G(x')$ to $R(x')$ according to the two losses in Eq. (10).

## 5   Conclusion

In this paper, we propose a VAE+classifier structure "class-disentangled variational autoencoder (CD-VAE)" to decompose an input image $x$ into $x = G(x) + R(x)$, where $R(x) = x - G(x)$ captures the essential information for classification while $G(x)$ covers all class-redundant information. We further propose an objective to jointly train the VAE and classifier to gain such class-disentanglement capability. CD-VAE provides novel perspectives to understand (1) how a neural net classifier predicts the class of an image and (2) how an adversarial example attacks the class. Inspired by these observations, we propose to conduct adversarial detection and adversarial defense on $R(x)$ and $G(x)$, respectively. These two simple strategies substantially improve the detection and defense against various adversarial attacks.

# 6 Acknowledgements

This work was supported in part by NSFC No. 61872329 and the Fundamental Research Funds for the Central Universitie sunder contract WK3490000005. We would like to thank NeurIPS area chairs and anonymous reviewers for their efforts in reviewing this paper and their constructive comments! We acknowledge the support of GPU cluster built by MCC Lab of Information Science and Technology Institution, USTC.

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
