# A    Class-Disentanglement results with different $\gamma$

**Labels:** Siamese cat, kelpie, Chesapeake bay retriever, Siberian husky, Curly coated retriever, Titi, Guenon, Bull mastiff

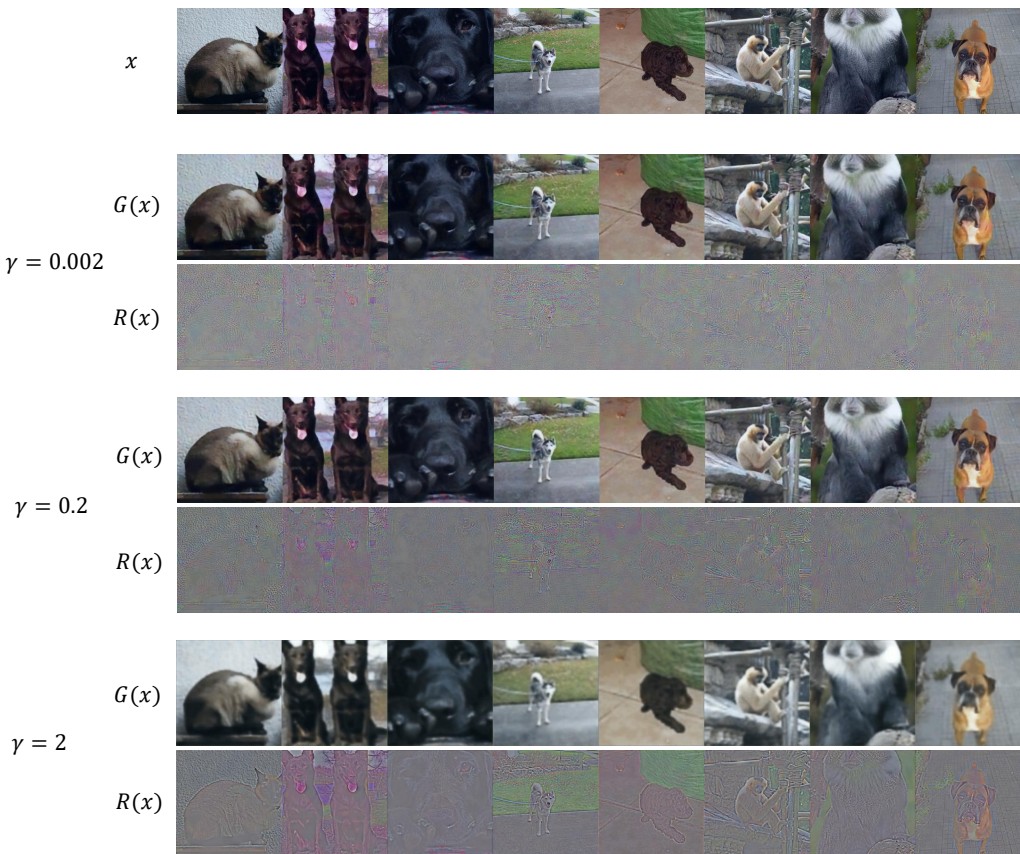

Figure 5: Class-Disentanglement results on restricted ImageNet by CD-VAE with different $\gamma$.

Changing $\gamma$ also leads to visual changes of $R(x)$ and $G(x)$ shown in Fig. 5, which provides interesting perspectives for interpretation of neural nets. As $\gamma$ increasing, $R(x)$ starts by first capturing the most discriminative (but sparse) features, e.g., the tongue/nose of dogs and the hair of monkeys, and gradually adds more class-relevant features. On the contrary, $G(x)$ becomes more blurry and lose more details but still tends to preserve the colors and shapes that are more critical to reconstruction.

# B    White-box Detection Performance

In this section, we evaluate the robustness of our method to detect adversarial images which may fool the detector in a white-box setting, i.e., one can have access to both the parameters of the classifier and the detector. We follow the setting of MD [30], i.e., Sec. E of [30], and use PGD attack to generate adversarial images, which maximizes the classification loss and minimizes the Mahalanobis distance at the same time. The results are given in Table 8: our method using $R(x)$ as input significantly outperforms the baseline using $x$ as input in the white-box setting.

|  | $\ell_\infty$ | | $\ell_2$ | |
|---|---|---|---|---|
|  | TNR | AUC | TNR | AUC |
| MD | 27.64 | 77.63 | 27.16 | 75.15 |
| MD($R(x)$) | 59.25 | 90.66 | 87.97 | 97.72 |

Table 8: White-box Detection Performance on CIFAR10.

# C    Detection Generalization Performance

In this section, we evaluate the generalizability of our detection method with other baselines. The detection models of the baselines and our method are trained using adversarial examples generated by FGSM attack. We then evaluate their generalizability by the detection AUC score against the other four unseen attacks, i.e., BIM, C&W, PGD-$l_\infty$ and PGD-$_2$. The hyper-parameters for attackers and architectures for detectors are the same as illustrated in Sec 4.2. Table that our method outperforms all three baselines by a large margin on detecting all the four unseen attacks, which demonstrates the generalizability of our detection strategy.

|  | BIM | C&W | PGD-$\ell_\infty$ | PGD-$\ell_2$ |
|---|---|---|---|---|
| KD | 94.82 | 94.75 | 94.59 | 93.62 |
| KD($R(x)$) | 97.86 | 96.89 | 97.95 | 98.20 |
| LID | 95.20 | 94.32 | 94.30 | 93.19 |
| LID($R(x)$) | 97.29 | 95.10 | 97.57 | 97.38 |
| MD | 96.13 | 96.05 | 96.34 | 92.37 |
| MD($R(x)$) | 99.21 | 99.13 | 99.26 | 99.13 |

Table 9: Detection Generalization Performance on CIFAR10. The detectors are trained on FGSM attack and evaluated on other four unseen attacks.

## D   Experimental Details of Adversarial Defense against White-Box Attacks

We train a pre-trained CD-VAE for 100 epoches using SGD optimizer with a initial learning rate of 1.0 and momentum of 0.9, where the learning rate is multiplied by 0.1 for every 30 epochs. We set $\gamma = 0.1$ and $\beta = 0.01$ in Eq.(8)-(10). We test baselines and our model against five attacks: $\ell_\infty$ and $\ell_2$ AutoAttack[3], JPEG [23], ReColor [28] and StADV [46]. AutoAttack is widely used for evaluation of robustness in recent works. It combines four strong attacks including two PGD variants and a black-box attack. JPEG attack generates perturbations in the frequency domain. ReColor uses a predefined function to recolor images. StAdv performs spatial transforms to images.

## E   Numerical Analysis of perturbation $\delta$ with respect to Larger $\epsilon$

In order to see the disentanglement effect of perturbation $\delta$ changes with respect to larger $\epsilon$, we try different values of $\epsilon$ (8/255, 48/255, 96/255, 144/255, 255/255) in PGD and report the $ell_p$-norm of $\delta$, $\delta_G$ and $\delta_R$ in Table 10. It shows that by increasing $\epsilon$, $G(x)$ suffers more distortion from the adversarial attacks. When $\epsilon$ is small (8/255 and 48/255), the $\ell_1$ and $\ell_2$ norm of is around 1/3 of that of $\delta$. When $\epsilon$ becomes large (144/255 and 255/255), the $\ell_1$ and $\ell_2$ norm of $\delta_G$ grows, exceeding 1/2 of that of $\delta$ . This demonstrates that with large $\epsilon$, the attackers first distort the class-essential information in $R(x)$ and then seek to perturb class-redundant information in $G(x)$. Note epsilon larger than 48/255 is rarely used for producing adversarial examples since it drastically changes the original images. So $G(X)$ is sufficiently robust to attacks using reasonable $\epsilon$ values.

## F   Convergence Curve

The objective in Eq. (1) is simple to optimize and the optimization converges fast after a few epochs. Thanks to the mutual information-bottleneck constraints between the VAE and the classifier, the VAE objective enforces the classifier to only pay attention to the most important information for classification, while the classifier's objective enforces the VAE to only reconstruct the class-redundant part. This helps to speed up the training of both models. Empirically, we do observe a fast and stable convergence on the VAE reconstruction accuracy, and the classifier's training accuracy in Fig. 6, within only 100 epochs (both models are trained from scratch on CIFAR10).

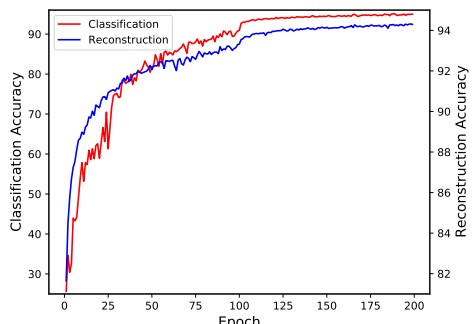

Figure 6: Convergence curve of CD-VAE.

[3]https://github.com/fra31/auto-attack

| $\epsilon$ | | $\ell_1 \times 10^{-3}$ | $\ell_2$ | $\ell_\infty$ |
|---|---|---|---|---|
| | $\delta$ | $13.65 \pm 0.88$ | $39.09 \pm 1.40$ | $0.14 \pm 0.00$ |
| 8/255 | $\delta_G$ | $4.15 \pm 0.65$ | $16.56 \pm 2.47$ | $0.39 \pm 0.17$ |
| | $\delta_R$ | $13.78 \pm 0.92$ | $40.97 \pm 1.87$ | $0.48 \pm 0.16$ |
| | $\delta$ | $70.10 \pm 4.04$ | $207.24 \pm 8.51$ | $0.84 \pm 0.00$ |
| 48/255 | $\delta_G$ | $19.67 \pm 3.20$ | $63.87 \pm 9.82$ | $0.90 \pm 0.43$ |
| | $\delta_R$ | $64.50 \pm 4.30$ | $192.35 \pm 10.47$ | $1.51 \pm 0.42$ |
| | $\delta$ | $125.91 \pm 7.41$ | $381.01 \pm 15.66$ | $1.68 \pm 0.00$ |
| 96/255 | $\delta_G$ | $49.67 \pm 9.73$ | $155.67 \pm 25.50$ | $2.74 \pm 0.32$ |
| | $\delta_R$ | $111.42 \pm 9.05$ | $337.48 \pm 23.83$ | $1.61 \pm 0.30$ |
| | $\delta$ | $173.23 \pm 10.86$ | $533.11 \pm 23.99$ | $2.52 \pm 0.00$ |
| 144/255 | $\delta_G$ | $88.40 \pm 15.69$ | $270.85 \pm 38.58$ | $2.36 \pm 0.20$ |
| | $\delta_R$ | $146.88 \pm 13.17$ | $446.07 \pm 34.35$ | $3.77 \pm 0.30$ |
| | $\delta$ | $256.76 \pm 28.93$ | $790.63 \pm 83.14$ | $4.43 \pm 0.07$ |
| 255/255 | $\delta_G$ | $141.29 \pm 33.16$ | $430.33 \pm 84.50$ | $3.30 \pm 0.25$ |
| | $\delta_R$ | $226.98 \pm 39.10$ | $653.33 \pm 87.96$ | $4.70 \pm 0.47$ |

Table 10: $\ell_p$ norm/distance of $\delta$ with respect to different $\epsilon$ (ImageNet).

## G  Additional Class-Disentanglement Results

In Fig. 2, we present the class-disentanglement results of several images from restricted ImageNet and their difference to the class-disentanglement results on the corresponding adversarial images. We do not show the adversarial images due to the space limit. In Fig. 7, we show the complete version of Fig. 2 with the adversarial images attached. In addition, we also present the similar results on CIFAR-10 in Fig. 8. It shows that the class-essential part only contains sparse and most discriminative features of an image, e.g., mouth of frog, wing of plane, etc, while $G(x)$ covers all the other redundant information for reconstruction. Moreover, the adversarial perturbation $\delta$ mainly exists in $\delta_R$.

In Table 2, for restricted ImageNet, we compare the $\ell_p$-norm of each class-disentangle component for both clean images and their adversarial images, as well as their differences in terms of $\ell_p$-distance, where $p \in \{1, 2, \infty\}$. In Table 11, we report similar results for CIFAR-10, which show consistent patterns as Table 2.

| | $\ell_1$ | $\ell_2$ | $\ell_\infty$ |
|---|---|---|---|
| $x$ | $2549.97 \pm 773.62$ | $53.06 \pm 14.68$ | $1.97 \pm 0.18$ |
| $G(x)$ | $2390.43 \pm 779.33$ | $49.56 \pm 14.71$ | $1.85 \pm 0.24$ |
| $R(x)$ | $537.87 \pm 124.00$ | $12.72 \pm 2.98$ | $1.13 \pm 0.27$ |
| $x'$ | $2545.17 \pm 761.72$ | $53.01 \pm 14.44$ | $2.00 \pm 0.16$ |
| $G(x')$ | $2382.31 \pm 774.14$ | $49.38 \pm 14.59$ | $1.84 \pm 0.23$ |
| $R(x')$ | $577.54 \pm 109.58$ | $13.45 \pm 2.76$ | $1.15 \pm 0.26$ |
| $\delta$ | $288.60 \pm 16.41$ | $5.60 \pm 0.47$ | $0.13 \pm 0.00$ |
| $\delta_G$ | $48.13 \pm 8.49$ | $1.11 \pm 0.19$ | $0.09 \pm 0.02$ |
| $\delta_R$ | $276.57 \pm 15.70$ | $5.42 \pm 0.46$ | $0.19 \pm 0.02$ |

Table 11: $\ell_p$ norm/distance of different parts in CD-VAE on CIFAR-10.

## H  Class-Disentanglement on Adversarially Trained Models

In Sec. 3.2 and G, we use CD-VAE to disentangle adversarial images generated by attacks against a classifier trained on clean data and illustrate that normally trained classifier mainly relies on class-essential part $R(x)$ for classification and adversarial perturbation $\delta$ mainly lies in $\delta_R$. In this section, we further use CD-VAE to disentangle the adversarial images generated by attacks against a robust

**Original label:** Vizsla, Macaw, Airedale, King crab, English springer, Gibbon, Weimaraner, Saluki

**Predicted label after attack:** Afghan hound, Lorikeet, Otterhound, Rock crab, Cocker spaniel, Patas, Chesapeake bay retriever, Basenji

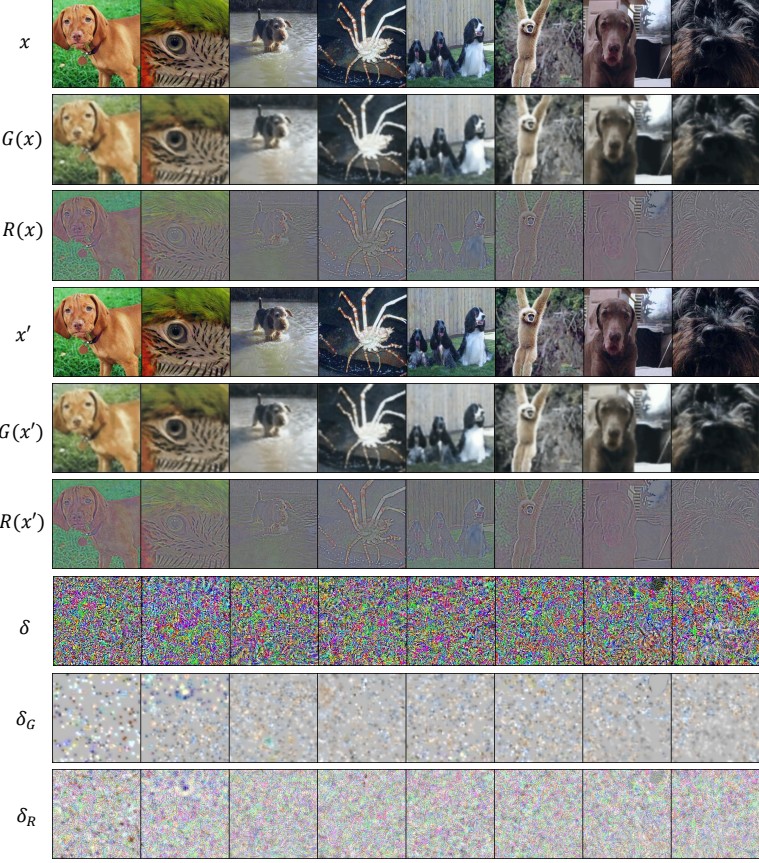

Figure 7: Class-Disentanglement results by CD-VAE on restricted ImageNet. The target model is a ResNet-50 trained on clean data of restricted ImageNet. The attack method is PGD-$\ell_\infty$ bounded with $\epsilon = 8/255$.

model (i.e., an adversarially trained model). It provides novel perspectives to understand how a robust model defend adversarial attacks.

We conduct the experiment on CIFAR-10. We use adversarial training [35] against PGD attack ($\ell_\infty$-ball constraint) to train a WideResNet-28-10 [48]. Then we generate adversarial images on the test set by PGD attack ($\ell_\infty$-ball constraint) towards this robust model. After that, we apply the trained CD-VAE model (the same model as used in Sec. 3.2 and G) to disentangle both adversarial images and clean images. We show the visualization of each disentangled part in Fig. 9. Comparing $\delta$, $\delta_R$ and $\delta_G$, we can find that $\delta$ has component on both $\delta_R$ and $\delta_G$. This indicates that the robust model relies on both $R(x)$ and $G(x)$ for classification, thus the attack has to cause change to both of them. This increases the difficulty of the attack, which can explain why the robust model can defend attacks.

In Table 12, we report the mean and standard deviation of $\ell_p$-norm for each class-disentangled component, where $p \in \{1, 2, \infty\}$. We can clearly see that $\delta_R$ and $\delta_G$ have comparable norm, which support the observation before.

# I  Disentanglement in Input-Space vs. Latent-Space

Latent-space disentanglement has been studied by many previous literatures [47, 18, 5, 10, 20, 24], but we are the first to perform class-disentanglement in pixel-space, which has different applications, formulation and conclusion compared with latent-space disentanglement.

**Original label:** Cat, Ship, Ship, Airplane, Frog, Frog, Automobile, Frog

**Predicted label after attack:** Frog, Automobile, Automobile, Ship, Deer, Cat, Horse, Cat

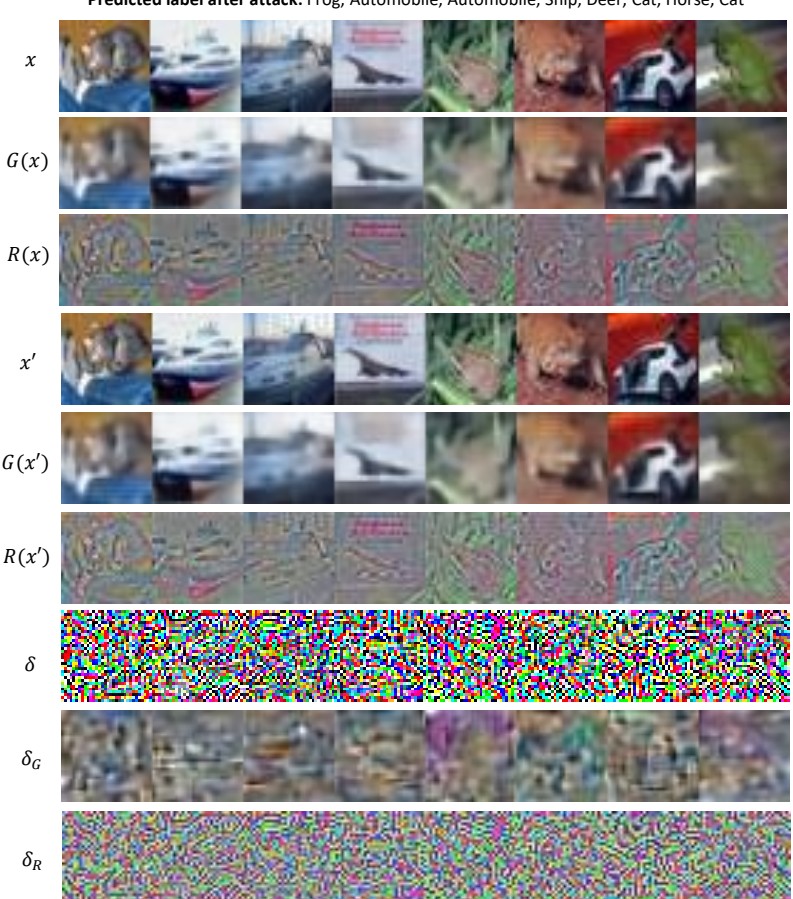

Figure 8: Class-Disentanglement results by CD-VAE on CIFAR-10. The Target model is a WideResNet-28-10 trained on clean data of CIFAR-10. The attack method is PGD-$\ell_\infty$ bounded with $\epsilon = 8/255$.

**Applications.** Our pixel-space disentanglement method can be applied to both adversarial detection and defense, while latent space disentanglement in latent space addresses either detection [47] or defense [18]. Moreover, our pixel-space disentanglement (e.g., $R(x)$ in Fig. 2 and Fig. 5)) provides a pixel-level interpretation tool for DNN classifiers and attacks against them, which leads to the empirical analysis in Sec. 3.2, while other latent-space disentanglement methods [47, 18] do not handle this problem. Our model produces a class-essential part $R(x)$ (input space), a class-redundant part $G(x)$ (input space), and a class-redundant representation $z$ (latent space) for any given input. Hence, our model can provide both image-like interpretation and low-dimensional abstract representations, while their model only provides the latter. With input-space class disentanglement, the class-essential and class-redundant information can be visualized as two images, while their latent counterparts are usually too abstract. Moreover, our method is complementary to and can be easily incorporated with existing methods of these tasks, e.g., by replacing their input $x$ with $G(x)$ or $R(x)$.

**Formulation.** Our method performs class-disentanglement by solving a simple unconstrained optimization in Eq. (1), which is easier and faster. Our mutual information-bottleneck constraints between the VAE and the classifier further speed up the training. On the contrary, [18] need to solve a constrained optimization problem to select filters and detects adversarial examples based on these selected filters, which is empirically difficult and expensive. [47] trains two feature extractors and two classifiers adversarially through a minimax Markov game with an objective composed of six loss functions, which is complicated and difficult to optimize.

**Original label:** Cat, Ship, Ship, Airplane, Frog, Frog, Automobile, Frog

**Predicted label after attack:** Dog, Ship, Automobile, Airplane, Cat, Frog, Automobile, Bird

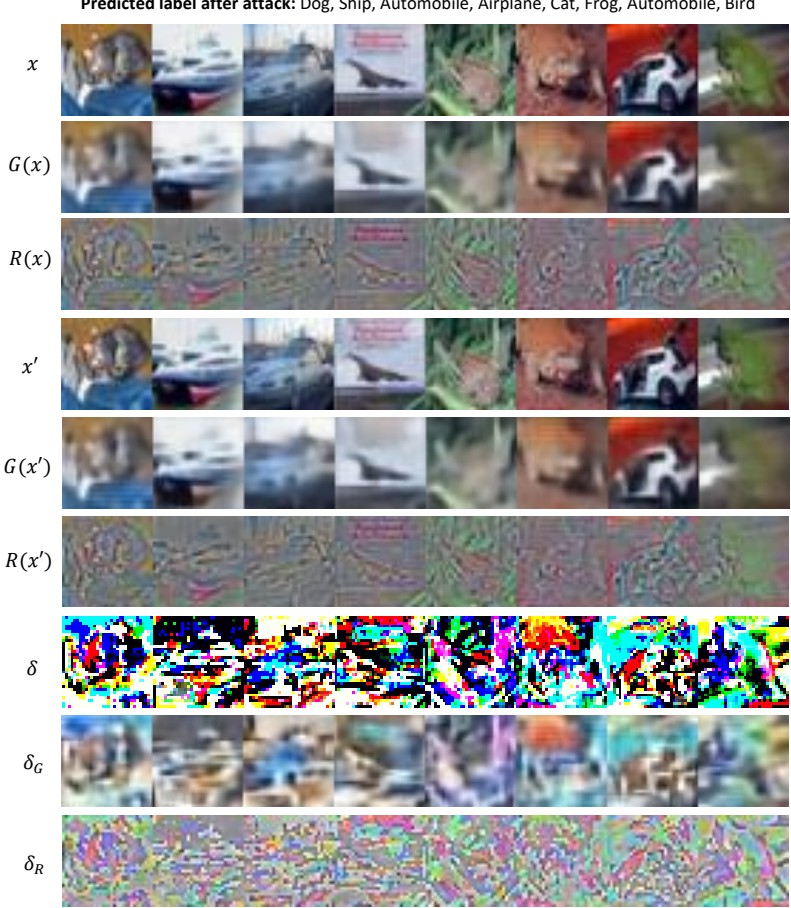

Figure 9: Class-Disentanglement of adversarial images generated by PGD-$\ell_\infty$ attack ($\epsilon = 8/255$) against an **adversarially trained** WideResNet-28-10 on CIFAR-10.

|  | $\ell_1$ | $\ell_2$ | $\ell_\infty$ |
|---|---|---|---|
| $x$ | $2521.73 \pm 808.00$ | $53.06 \pm 14.68$ | $1.97 \pm 0.18$ |
| $G(x)$ | $2363.96 \pm 808.70$ | $49.56 \pm 14.71$ | $1.85 \pm 0.24$ |
| $R(x)$ | $531.91 \pm 132.02$ | $12.72 \pm 2.98$ | $1.13 \pm 0.27$ |
| $x'$ | $2524.33 \pm 800.29$ | $53.14 \pm 14.53$ | $2.00 \pm 0.17$ |
| $G(x')$ | $2364.47 \pm 805.04$ | $49.57 \pm 14.63$ | $1.85 \pm 0.24$ |
| $R(x')$ | $553.87 \pm 128.40$ | $13.15 \pm 2.87$ | $1.15 \pm 0.27$ |
| $\delta$ | $354.70 \pm 38.53$ | $6.57 \pm 0.51$ | $0.13 \pm 0.00$ |
| $\delta_G$ | $190.65 \pm 28.74$ | $4.14 \pm 0.50$ | $0.23 \pm 0.03$ |
| $\delta_R$ | $214.76 \pm 28.23$ | $4.68 \pm 0.46$ | $0.26 \pm 0.02$ |

Table 12: $\ell_p$ norm/distance of different parts in CD-VAE for a robust model on CIFAR-10.

**Conclusion.** [47] detects adversarial perturbation in the class-irrelevant part, while our method avoids doing so since our study suggests that the adversarial perturbation mainly affects the class-dependent part, on which we instead conduct our adversarial detection. This also implies that class-disentanglement in the input space and latent space can have very different properties.