# OpenReview forum: "Class-Disentanglement and Applications in Adversarial Detection and Defense"
_NeurIPS.cc/2021/Conference — NeurIPS 2021 Poster_

### Official Review · Reviewer_81hb · 2021-07-15

**Rating:** 4
**Confidence:** 5

**Summary:**

This paper proposes a novel method called “class-disentanglement” for adversarial defense and detection. It decomposes one image into two parts: the reconstruction part, which contains the redundant information, and the residual part, which captures the essential information for prediction. Empirical results show that adversarial perturbations majorly lie in the residual part, and the reconstruction part still contains class information irrelevant to the residual part. Motivated by this observation, the authors improve existing adversarial detection methods, such as KD, LID, and MD, by implementing these methods on the residual part. Also, they train a classifier over the reconstruction part and exhibit competitive robustness.

**Main Review:**

Originality: This paper contributes some new ideas. But analyzing adversarial examples with class-disentanglement is not so novel, such as the work [1], which considers this problem in the latent space.
Quality: This paper is well organized.
Clarity: The paper is easy to read. However, there still are some typos.
Significance: This paper reveals an interesting observation that the information for classification is redundant. Adversarial perturbation merely impacts a fraction of them. However, the experimental results are not sufficient.

Weaknesses and my questions of this paper:
1.	I suspect the cosine similarity report in Table 1. The mean and standard deviation of the cosine similarity between G(x) and G(x’) is 1 and 0, respectively. That means, for all considered x, G(x) and G(x’) are exactly the same up to a constant coefficient. It seems abnormal.

2.	In Table 2, the author report Lp norms of δ and δR to support their claim that “ the adversarial perturbation δ mainly lies in δR…” (lines 198-202). However, even if δ and δR have similar Lp norms, possibly we cannot conclude that they are correlated.

3.	Generalizability is quite important to evaluate the effectiveness of adversarial detection or defense methods. However, the authors merely considered the case where the adversarial examples for training and evaluating the detector or defense model are generated by the same attack method.
Other limitations:
1.	The line 222 has a typo “ In order to”
2.	The line 292 has a typo “ sImageNet”

[1] Yang, Shuo et al, “Adversarial Robustness through Disentangled Representations”, Proceedings of the AAAI Conference on Artificial Intelligence, 2021.

**Time Spent Reviewing:**

2 hours

---

> ### Author Response · Authors · 2021-08-10
> **Response to Reviewer 81hb:**
>
> We appreciate your time and suggestions! As you suggested, we have added an experiment evaluating the generalizability of our detection method to different types of adversarial attacks. Here are our detailed replies to your questions.
>
> **(1) Analyzing adversarial examples with class-disentanglement is not so novel, such as the work [1], which considers this problem in the latent space.**
>
> Our method differs from [1] in several fundamental aspects:
>
> (i) **The mathematical formulation and network architecture of [1] is very different from ours**. [1] trains two feature extractors and two classifiers adversarially through a minimax Markov game with an objective composed of six loss functions, which is **complicated and difficult to optimize**. While **both our optimization problem (minimizing VAE loss + classification loss) and our network architecture (VAE + classifier) are much simpler**. For the input, their model takes two inputs, i.e., a clean example $x$ and its adversarial example $x'$, while our model only takes one of them ($x$ or $x'$). For the output, they produce class-specific and class-irrelevant representations in a latent space for the two inputs, while our model produces a class-essential part $R()$ (input space), a class-redundant part $G()$ (input space), and a class-redundant representation $z$ (latent space) for any given input. Hence, our model can provide both image-like interpretation and low-dimensional abstract representations, while their model only provides the latter. With input-space class disentanglement, the class-essential and class-redundant information can be visualized as two images, while their latent counterparts are usually too abstract.
>
> (ii) **The main conclusion of [1] differs from ours: [1] detects adversarial perturbation in the class-irrelevant part, while our method avoids doing so** since our study suggests that the adversarial perturbation mainly affects the class-dependent part, on which we instead conduct our adversarial detection. This also implies that **class-disentanglement in the input space and latent space can have very different properties**.
>
> (iii) **Class-disentanglement in the input space (ours) has different applications to the latent space [1] disentanglement**. For example, our method can be applied to both adversarial detection and defense, while the latent space disentanglement in [1] only addresses the latter; our disentanglement (e.g., $R(x)$ in Fig. 2 and Fig. 5) provides a pixel-level interpretation tool for DNN classifiers and attacks against them, which leads to the empirical analysis in Sec. 3.2, while [1] does not address this problem. On the other hand, our model can potentially be applied to many latent space applications since the bottleneck features $z$ in CD-VAE are class-disentangled latent representations, e.g., $z$ can be used to train a robust classifier.
>
> We hope the above can convince you of our fundamental difference and main novelty compared to [1]. For empirical comparisons to [1], we contacted the authors for the code of [1] but they cannot provide the code right now. We will run an experimental comparison and post the results if they provide the code before the closure of the discussion phase.
>
> **(2) I suspect the cosine similarity report in Table 1. The mean and standard deviation of the cosine similarity between $G(x)$ and $G(x’)$ is 1 and 0, respectively. It seems abnormal.**
>
> This is normal due to the **tiny perturbations of adversarial attacks**. In almost all adversarial attacks [2-5], to make the attacks unnoticeable for humans, the adversarial example $x'$ is constrained to reside within a small $\epsilon$-ball around the original example $x$ ($\|x-x'\|_\infty\leq0.14$ in the experiments). Since the VAE objective in Eq. (2) aims to reconstruct $x$ by $G(x)$, $G(x)$ and $G(x')$ are also close to each other with cosine similarity of nearly 1 (the mean and std are not exactly $1$ and $0$ but $>0.99$ and $<0.01$). To see this, **we make the cosine similarity accurate to four decimal places** in them and add the comparison between $x$ and $x'$ for reference:
>
> ||Cosine Similarity|FID|
> |:--|:--|:--|
> |$x$ vs. $x'$| 0.9945$\pm$0.0028 | 10.99 |
> |$G(x)$ vs. $G(x')$|  0.9997$\pm$0.0003 | 0.90 |
> |$R(x)$ vs. $R(x')$|  0.9460$\pm$0.0260 | 22.31 |
>
> In the table above, the cosine similarity between $x$ and $x'$ is also very large (0.9945$\pm$0.0028), and the cosine similarity between $G(x)$ and $G(x')$ is slightly larger than that between $x$ and $x'$. Moreover, the Frechet Inception Distance between $G(x)$ and $G(x')$ is smaller than that between $x$ and $x'$ and that between $R(x)$ and $R(x')$. This is another evidence verifying that $G(x)$ and $G(x')$ are truly close.
>
> **(3) In Table 2, the author report $\ell _ p$ norms of $\delta$ and $\delta _ R$ to support their claim that the adversarial perturbation $\delta$ mainly lies in $\delta _ R$ (lines 198-202). However, even if $\delta$ and $\delta _ R$ have similar $\ell _ p$ norms, possibly we cannot conclude that they are correlated.**
>
> Only comparing the $\ell _ p$ norm of $\delta$ and $\delta _ R$ is insufficient to reach our conclusion. **One also needs to look at the $\ell _ p$ norm of $\delta _ G = \delta - \delta _ R$ in Table 2**, which is much smaller than that of $\delta$ and $\delta _ R$ (except the $\ell _ \infty$ norm since it only takes the maximal element). Hence, we can conclude that $\delta$ and $\delta _ R$ are close to each other and are strongly correlated.
>
> **(4) Generalizability is quite important to evaluate the effectiveness of adversarial detection or defense methods. However, the authors merely considered the case where the adversarial examples for training and evaluating the detector or defense model are generated by the same attack method.**
>
> (i) **For adversarial defense, we have shown the generalizability of our method in Table 7**. Our model trained on C\&W-$\ell _ \infty$ attacks does not only exhibit robustness against the $l _ \infty$-norm bounded AutoAttack but also generalizes well to other four types of unseen attacks, i.e., $l _ 2$-norm bounded AutoAttack, JPEG, ReColour and StADV. In Table 7, The generalizability to unseen attacks is measured by "Unseen Attacks (mean)", the defense accuracy averaged over all the attacks not used for the adversarial training of the defense model. Our method achieves the highest (51.4\%) "Unseen Attacks (mean)" among all the compared methods.
>
> (ii) **For adversarial detection, in the table below, we compare the generalizability of our detection method with other baselines** on CIFAR-10. The detection models of the baselines and our method are trained using adversarial examples generated by FGSM attack. We then evaluate their generalizability by the detection AUC score against the other four unseen attacks, i.e., BIM, C\&W, PGD-$\ell _ \infty$ and PGD-$\ell _ 2$. The hyperparameters for attackers and architectures for detectors are the same as illustrated in Sec 4.2.  We can see that our method outperforms all three baselines by a large margin on detecting all the four unseen attacks, which demonstrates the generalizability of our detection strategy.
>
> |Method|BIM|C\&W|PGD-$\ell _ \infty$|PGD-$\ell _ 2$|
> |:------|:--|:--|:--|:--|
> |KD| 94.82 | 94.75|94.59|93.62|
> |KD ($R(x)$)|  97.86 |96.89 |97.95|98.20|
> |LID| 95.20 | 94.32|94.30|93.19|
> |LID ($R(x)$)| 97.29 | 95.10|97.57|97.38|
> |MD| 96.13 | 96.05|96.34|92.37|
> |MD ($R(x)$)|  99.21 |99.13 |99.26|99.13|
>
> **(5) The paper is easy to read. However, there still are some typos.**
>
> Thanks for your suggestion! We will fix those typos.
>
> ***
>
> [1] Yang, Shuo et al. Adversarial Robustness through Disentangled Representations. AAAI, 2021.
>
> [2] Leslie Rice, Eric Wong, J. Zico Kolter. Overfitting in adversarially robust deep learning. ICML, 2020.
>
> [3] Lang Huang, Chao Zhang, Hongyang Zhang. Self-Adaptive Training: beyond Empirical Risk Minimization. ICML 2020.
>
> [4] Cassidy Laidlaw, Sahil Singla, Soheil Feizi. PERCEPTUAL ADVERSARIAL ROBUSTNESS: DEFENSE AGAINST UNSEEN THREAT MODELS. ICLR 2021
>
> [5] Hongyang Zhang, Yaodong Yu, Jiantao Jiao, Eric P. Xing, Laurent El Ghaoui, Michael I. Jordan. Theoretically Principled Trade-off between Robustness and Accuracy. ICML, 2019.

---

> ### Author Response · Authors · 2021-08-27
> **All your concerns are addressed with new experimental evidences. Your two suspects do not hold. Would you mind checking our response? Thanks!**
>
> Dear Reviewer 81hb:
>
> We appreciate your comments and time! We addressed all your concerns with a **detailed response below**. Would you mind checking it and confirm if you have further questions? Here is a short summary:
>
> * For originality, we list **three fundamental differences of our method compared to [1]** you mentioned. We can empirically compare with it when its code is available.
>
> * For 1-3 in your questions, we provide evidence from our paper and new experiments to show that **your suspects about (1) the cosine similarity in Table 1 and (2) the correlation between $\delta$ and $\delta_R$ in Table 2 do Not hold**. We will remove the typos pointed out by 4-5 of your questions.
>
> Would you mind letting us know if you have any further concerns? Thanks a lot!
>
> Best Regards,
>
> Authors

---

> ### Author Response · Authors · 2021-09-01
> **[Last two days reminder] We addressed all your concerns with new experiments. Would you mind confirming if you have further concerns and reconsidering your rating? Thanks!**
>
> Dear Reviewer 81hb:
>
> We appreciate your comments and time! We have addressed all your concerns with a short summary and a detailed response below. We also reported new experimental results according to your main concerns: they show that the suspects in your original comments do not hold. Since we only have two days left for the discussion, would you mind checking it and confirm if you have further questions? If not, we humbly expect that you can reconsider the rating? Thanks a lot!
>
> Best Regards,
>
> Authors

---

### Official Review · Reviewer_mKY1 · 2021-07-16

**Rating:** 7
**Confidence:** 4

**Summary:**

Use VAEs to extract "class-dependent information". Validate that adversarial perturbations are more prone to lie in the class-dependent component of the decomposition. Leverage the decomposition for detection and defense.

**Limitations And Societal Impact:**

I would not say the authors adequately addressed the limitations and potential negative societal impact of their work. While I don't see any unique potential negative societal impact of this work, a extended, clear limitations discussion focused on the reduction in clean accuracy, could have, in part, ameliorated my concerns with regards to this paper.

**Main Review:**

*Originality*: Existing work which utilized VAEs for adversarial defense does not appear to be cited (https://arxiv.org/abs/1805.09190, https://arxiv.org/abs/1802.06552, https://arxiv.org/abs/1806.00081, https://arxiv.org/abs/1906.00230).

Furthermore, I would be surprised that the idea of decomposing an image into class-independent and class-dependent features has not been explored before in, say, the "interpretability" literature, however the related work solely focuses on generic methods for disentangled representation learning, adversarial detection, and adversarial defense. I would have much appreciated the related work section would focus on, or at least have a section regarding, the novelty of the task-dependent decomposition.

*Quality*: I am wary that the work relies upon suboptimality in optimization, which is reflected in a significant reduction in clean accuracy visible in Table 6. Given this, I would not say this work provides a sound contribution for detection/defense where the performance bounds are well-understood for practitioners.

*Clarity*: I would not say the submission is clearly written, and thereby I would not say the work is easily reproducible. More details below.

Could the authors clarify how the adversarial was generated in Sec. 3.2? Equation (1) implies D() is trained on x - G(x), I am not sure why then D() should classify x correctly, which is the presumption with which standard adversarial attacks, i.e. what was detailed in Eq. (4) rests upon.

On this point, it does not appear a D() is trained jointly with G() with regards to the experiment specified in Sec. 3.2, and thus going forward in the experiments section, it is not clear whether a) D() is pretrained on the input images and fixed/frozen when optimizing G(), or b) D() is trained simultaneously with G(), as Eq. (1) implies. Regardless, the reader should not have to make any assumptions regarding such significant details.

Regarding Sec. 3.4, if again the above presumptions are correct, G() is optimized s.t. it can reconstruct the given image well, while if subtracted from the input, the pretrained classifier functions well. This trade-off should lead to G() prioritizing capturing class-independent features, as mentioned by the authors. More specifically, G() should capture as many features as possible, for improved reconstruction, thus only the minimal necessary features for the classifier to function should be discarded. If G() is optimal, then yes the output of G() intuitively should be less perturbed by an adversarial, but it is questionable whether the output of G() should be useful for classification, given what the classifier *needs* to be accurate, has been by definition, removed. Thus, why classifying the output of G() should work, seems to rely on a somewhat mysterious interplay in the learning objective trade-off, and this lack of clarity makes me question the robustness of the demonstrated empirical results in the work.

In line with the above argument, a significant reduction in the clean accuracy can be seen in Table 6, however this clear detriment is not discussed by the authors.

*Significance*: a) not clear that the approach is novel given the lack of discussion in related work on class-dependent decomposition of images; b) not clear if this work can be built upon given critical missing details in the main paper, namely with regards to the optimization of D(); c) not clear, *why* the defense in particular works, and as it currently stands, using the defense leads to a significant reduction in clean accuracy.

**Post-Rebuttal**: Authors have addressed my concerns in full, and I have thus raised my score.

**Time Spent Reviewing:**

2-3 hours

---

> ### Author Response · Authors · 2021-08-10
> **Response to Reviewer mKY1:**
>
> We appreciate your time and suggestions! Here are our detailed replies to your questions.
>
> **(1) Existing work which utilized VAEs for adversarial defense does not appear to be cited (https://arxiv.org/abs/1805.09190, https://arxiv.org/abs/1802.06552, https://arxiv.org/abs/1806.00081, https://arxiv.org/abs/1906.00230).**
>
> We will discuss and compare these works in detail. Here is a summary. The main difference is that we jointly train VAE with a classifier for class-disentanglement, whose results can explain how and why adversarial attacks work in the input space. In contrast,  the mentioned papers only train a VAE and do not address the class-disentanglement problem. Moreover, the class-disentanglement property enables us to improve both adversarial detection and defense using the same model, while each mentioned paper merely tackles one of these two problems, e.g., adversarial detection only (https://arxiv.org/pdf/1802.06552) or defense only (https://arxiv.org/abs/1805.09190, https://arxiv.org/abs/1806.00081, https://arxiv.org/abs/1906.00230).
>
> **(2) Furthermore, I would be surprised that the idea of decomposing an image into class-independent and class-dependent features has not been explored before in, say, the "interpretability" literature, however the related work solely focuses on generic methods for disentangled representation learning, adversarial detection, and adversarial defense.**
>
> To the best of our knowledge, we are the first to decompose an image as the sum of a class-independent part and a class-dependent part in the **input space**. In L90 of Sec.2 (Related Work), we have discussed some related task-dependent decomposition methods decomposing the content and style of an image [1-4]. Probably the most related works are the class-disentangle methods from [5] and [6]. However, they perform class-disentanglement in the **latent space**, causing significant differences in their optimization formulations, model architectures, and applications. In "interpretability" literature, the interpretation is usually derived for a pre-trained classifier and only reflects its properties, while our method reveals the general properties of DNNs (independent of our CD-VAE model) and a rich class of adversarial attacks.
>
> **(3) Could the authors clarify how the adversarial was generated in Sec. 3.2?**
>
> We follow the widely used routine to generate the adversarial examples: The classifier $D(\cdot )$ being attacked in Sec. 3.2 is pre-trained (independently) on clean data until convergence. Then we apply $\ell _ \infty$ PGD attack as defined in Eq. (4) to generate adversarial examples. Note the attacked $D(\cdot)$ here is different from the $D(\cdot)$ in Eq. (1) of CD-VAE, as clarified in L184. It is neither jointly trained with CD-VAE. The classifier in CD-VAE is only used to extract the class-essential information and performs as an information bottleneck for the VAE.
>
> **(4) I am wary that the work relies upon suboptimality in optimization, which is reflected in a significant reduction in clean accuracy visible in Table 6.**
>
> (i) **The accuracy-robustness trade-off is a common phenomenon observed on most adversarial defense methods** [7-10]. In Table 6, all the defense methods suffer from different levels of accuracy degradation comparing to ''Normal''. Our method achieves superior robustness (e.g., over 77\% for $\ell _ \infty$ attack on CIFAR-10) across all types of attacks and its clean data accuracy (86.81\%) is comparable to other state-of-the-art defense methods.
>
> (ii) The accuracy drop for our method is due to the removal of class-essential information $R(x)$, which contains the most useful information for classification but is also vulnerable to adversarial attacks, as demonstrated in the empirical study of Sec. 3.2.
>
> (iii) We do not suffer from suboptimality. Thanks to the objective in Eq. (1), where the VAE and the classifier form mutual information-bottleneck constraints for each other, the class-essential and class-redundant information are quickly distributed to $R(x)=x-G(x)$ and $G(x)$ respectively after a few epochs. For example, our training converges very fast within only 60 epochs on CIFAR10, and here is the training loss sequence as well as the VAE reconstruction accuracy and the classifier's training accuracy:
>
> |Epoch:|10|20|30|40|50|60|70|
> |:------|:--|:--|:--|:--|:--|:--|:--|
> |$L_G$| 1.12 | 0.96|0.87|0.82|0.77|0.75|0.76|
> |$L_D$|  0.93 |0.66 |0.53|0.46|0.40|0.38|0.39|
> |$G(x)$ Reconstruction Accuracy (\%)|90.33|92.0|92.7|93.2|93.8|93.9|93.8|
> |$R(x)$ Classification Accuracy (\%)|67.27|76.7|81.3|83.3|85.9|86.5|85.9|
>
> **(5) Why classifying the output of G() should work, seems to rely on a somewhat mysterious interplay in the learning objective trade-off, and this lack of clarity makes me question the robustness of the demonstrated empirical results in the work.**
>
> In Sec. 4.1, we provided an empirical analysis of the classification-reconstruction trade-off via tuning $\gamma$. In Table 3 and the analysis below it, we explained why $G(x)$ can be less perturbed by an adversarial and meanwhile useful to classification: because the class-relevant information preserved in $G(x)$ is very different to that being mainly distorted by adversarial attacks.
>
> One can look at the test accuracy of a classifier trained on $G(x)$ but tested on $x$, $R(x)$, or $G(x)$: the accuracy is low if the classifier takes $x$ or $R(x)$ as input but is high when taking $G(x)$ as input. It indicates that $G(x)$ does preserve class-relevant information (because of the high test accuracy on $G(x)$) but this information is orthogonal to and very different from the class-relevant information the attacked classifiers (trained on $x$) heavily rely on, where the latter is the main focus of adversarial attacks since the attacks are optimized to degenerate the classifier trained on $x$.
>
> ***
>
> [1] Diane Bouchacourt, Ryota Tomioka, and Sebastian Nowozin. Multi-level variational autoencoder: Learning disentangled representations from grouped observations. AAAI, 2018.
>
> [2] Abel Gonzalez-Garcia, Joost Van De Weijer, and Yoshua Bengio .Image-to-image translation
> for cross-domain disentanglement. arXiv:1805.09730, 2018.
>
> [3] Maximilian Ilse, Jakub M Tomczak, Christos Louizos, and Max Welling. Diva. Domain invariant variational autoencoders. PMLR, 2020.
>
> [4] Yen-Cheng Liu, Yu-Ying Yeh, Tzu-Chien Fu, Sheng-De Wang, Wei-Chen Chiu, and Yu-Chiang Frank Wang. Detach and adapt: Learning cross-domain disentangled deep representation. CVPR, 2018.
>
> [5] Yang, Shuo et al. Adversarial Robustness through Disentangled Representations. AAAI, 2021.
>
> [6] Sina Hajimiri, Aryo Lotfi, Mahdieh Soleymani Baghshah. Semi-Supervised Disentanglement of Class-Related and Class-Independent Factors in VAE. arXiv:2102.00892.
>
> [7] Hongyang Zhang, Yaodong Yu, Jiantao Jiao, Eric P. Xing, Laurent El Ghaoui, Michael I. Jordan. Theoretically Principled Trade-off between Robustness and Accuracy. ICML, 2019.
>
> [8] Leslie Rice, Eric Wong, J. Zico Kolter. Overfitting in adversarially robust deep learning. ICML, 2020.
>
> [9] Lang Huang, Chao Zhang, Hongyang Zhang. Self-Adaptive Training: beyond Empirical Risk Minimization. ICML 2020.
>
> [10] Cassidy Laidlaw, Sahil Singla, Soheil Feizi. PERCEPTUAL ADVERSARIAL ROBUSTNESS: DEFENSE AGAINST UNSEEN THREAT MODELS. ICLR 2021.

---

> > ### Comment · Reviewer_mKY1 · 2021-08-22
> > **Response**
> >
> > I thank the reviewers for their rebuttal. I encourage the authors to include the explanations provided in the revision, so the experiments are fully clear to the reader.
> >
> > Upon the authors' clarifications, **why** the method should work is now clear to me; if there exist imperceptible features which are correlated with the class label, said features should be included in $R(x)=x-G(x)$. This, since adversarial perturbations are imperceptible by design, an optimal $G(\cdot)$ should discard it.
> >
> > In the adversarial detection and adversarial defense experiments, it appears the adversarial attacks targeted $D(\cdot)$. It should be noted however that the evaluation as a defense and a detector is currently incomplete, as it appears that the attacks were not white-box attacks, let alone perfect-knowledge attacks.
> >
> > For detection, the authors follow previous work in building a KDE, but apply the method to $R(x)$. However, the attack is unaware of the detection mechanism, from the attacker's perspective, test-time inference is conducted by querying $D(\cdot)$ with the query input $x$, not i) querying $D(\cdot)$ with $R(x)$ and ii) checking if a trained kernel density is below a threshold. As discussed in prior work (https://arxiv.org/abs/2002.08347), the attacks used were generic, they did not target the KDE, and thus do not constitute adaptive attacks. Crucially, the attack was unaware of $G(\cdot)$.
> >
> > For defense, this is very clearly not a white-box evaluation, as the attacker attempts to fool the pretrained network, but the defender preprocesses the adversarial example with $G(\cdot)$, which again the attacker was unaware of.
> >
> > In short, the authors did not evaluate whether the defense was vulnerable if the attacker had access to $G(\cdot)$, and did not design an adaptive attack which attacked the inference actually done in practice, which additionally involves a built KDE in the case of detection.
> >
> > This observation is more striking when one observes that in Table 4, that the baselines were exposed to the attack, as the sole distinction between the baselines and the proposition is the use of $R(x)$ by the proposition as a preprocessing. In Table 5, the attacks constitute white-box attacks against the baselines, as the adversarial example was generated under the same inference strategy as is used for testing, but in the case of the proposed model, the attack targets the pretrained model, without knowledge of $G(x)$ being used as a preprocessing.
> >
> > Altogether, the empirical comparison does not appear fair to prior work as it currently stands, and thus does not demonstrate the empirical benefit of the proposed strategy relative to prior work. I suggest the authors precisely specify their threat model, i.e. what the attack is assumed to "know"/have access to, and what the attack is assumed not to "know"/have access to. With the threat model specified, all baseline comparisons should take place under the same threat model in order for there to be a fair comparison.
> >
> > In revising the experiments, I would suggest the authors take a look at https://arxiv.org/abs/1902.06705 and the aforementioned https://arxiv.org/abs/2002.08347, and check whether the evaluation adheres to the suggestions made by said works.

---

> > > ### Author Response · Authors · 2021-08-24
> > > **Response to Reviewer mKY1: Our comparisons are fair and we do report white-box evaluations**
> > >
> > > We are glad to hear that our method is now clear to the reviewer. We appreciate the reviewer for the concerns regarding fair comparisons and white-box evaluations! **However, we believe that the reviewer still misread/misinterpret some parts of our experiments. Our empirical comparisons are fair to all methods and we do evaluate them against the most challenging white-box attacks** that can access every part of our CD-VAE model (Section A of Appendix). In different settings, our method consistently outperforms most baselines across all evaluated attacks. Here are our detailed reply to the concerns.
> > >
> > > **(1) For defense, this is very clearly not a white-box evaluation, as the attacker attempts to fool the pretrained network, but the defender preprocesses the adversarial example with $G(x)$, which again the attacker was unaware of.**
> > >
> > > We did compare the defense performance of different methods under **white-box attacks in Table 7 of Appendix**. In this setting, the attacker can access every part of the defense model, including $G(\cdot)$ of our CD-VAE model, and we train our CD-VAE model using a modified adversarial training objective in Eq. (8)-(10). We evaluate all defense methods against five white-box attacks, i.e., $\ell _2$ and $\ell _ \infty$-bounded AutoAttack, JPEG, Recolor and StAdv. In Table 7, our method achieves the highest defense accuracy over "Unseen Attacks (mean)", which is the defense accuracy averaged over all the attacks excluding the one used for adversarial training, and **our defense accuracy is >10\% higher than the best baseline**. It also achieves the highest defense accuracy against two out of the five white-box attacks, i.e., JPEG and ReColor.
> > >
> > > **The comparisons in Table 6 and Table 7 are FAIR**, though each focuses on a different defense setting. In Table 6, attackers can access the classification models of all the methods but cannot access the data preprocessing models (if any). In Table 7, attackers can access every part of the defense model including the data preprocessing parts, e.g., $G(\cdot)$ of our CD-VAE model. In both settings, our method outperforms other baselines by a large margin across diverse types of attacks.
> > >
> > > **(2) For detection, the authors follow previous work in building a KDE, but apply the method to $R(x)$. However, the attack is unaware of the detection mechanism, from the attacker's perspective, test-time inference is conducted by querying $D(\cdot)$ with the query input $x$, not i) querying $D(\cdot)$ with $R(x)$ and ii) checking if a trained kernel density is below a threshold. As discussed in prior work (https://arxiv.org/abs/2002.08347), the attacks used were generic, they did not target the KDE, and thus do not constitute adaptive attacks. Crucially, the attack was unaware of $G(\cdot)$.**
> > >
> > > (i) "test-time inference is conducted by querying $D(\cdot)$ with the query input $x$, not i) querying $D(\cdot)$ with $R(x)$" --- **This is incorrect!** When evaluating our detection strategy, the attackers generate adversarial examples by **querying $D(\cdot)$ with $R(x)$**. Therefore, the attackers mainly distort $R(x)$, making our strategy of detecting the attacks on $R(x)$ more effective. Hence, **the attacks are clearly aware of $G(\cdot)$ (since $R(x)=x-G(x)$) and the comparison on the adversarial detection task is fair.**
> > >
> > > (ii) Many mainstream detection methods (e.g., KD[1] and LID[2]) are not designed for white-box attacks so they have not been evaluated under adaptive attacks in their papers. Hence, in the paper, we mainly show that **our simple detection strategy can improve those methods in the same settings used in their original papers**, which is a noticeable contribution.
> > >
> > > (iii) **In fact, our detection strategy can bring even more improvements under adaptive/white-box attacks**, which target at both the classifier and the KDE. This is shown in the table below: our method using $R(x)$ as input significantly outperforms the baseline using $x$ as input in the white-box setting.
> > > In particular, we apply our strategy to MD[3] because MD[3] considers adaptive attacks in their paper. We follow their setting, i.e., Sec. E of [3], and use PGD attack to generate adversarial images $x'$, which maximizes the classification loss and minimizes the Mahalanobis distance at the same time. The detection accuracy on CIFAR-10 are reported in the table below and the features being attacked concatenate the outputs of all the residual blocks in WideResNet.
> > >
> > > ||$\ell _ \infty$||$\ell _2$||
> > > |:------|:--|:--|:--|:--|
> > > || TNR | AUC|TNR|AUC|
> > > |MD|  27.64 |77.63 |27.16|75.15|
> > > |MD($R(x)$) |**59.25**|**90.66**|**87.97**|**97.72**|
> > >
> > > **(3) Altogether, the empirical comparison does not appear fair to prior work as it currently stands, and thus does not demonstrate the empirical benefit of the proposed strategy relative to prior work.**
> > >
> > > Due to the above reasons, **this criticism is groundless and we respectfully disagree with it**. In both defense and detection tasks, under both grey-box and white-box attacks, our comparisons are fair and consistently demonstrate the advantages of our method over other baselines.
> > >
> > > ---
> > > [1] Reuben Feinman, Ryan R. Curtin, Saurabh Shintre, Andrew B. Gardner. Detecting Adversarial Samples from Artifacts. arxiv:1703.00410, 2017.
> > >
> > > [2]Xingjun Ma, Bo Li, Yisen Wang, Sarah M. Erfani, Sudanthi Wijewickrema, Grant Schoenebeck, Dawn Song, Michael E. Houle, James Bailey. CHARACTERIZING ADVERSARIAL SUBSPACES USING LOCAL INTRINSIC DIMENSIONALITY. ICLR, 2018.
> > >
> > > [3] Kimin Lee, Kibok Lee, Honglak Lee, Jinwoo Shin. A Simple Unified Framework for Detecting Out-of-Distribution Samples and Adversarial Attacks. NeurIPS, 2021

---

> > > > ### Comment · Reviewer_mKY1 · 2021-08-24
> > > > **Response**
> > > >
> > > > Thanks for pointing me to Table 7 in the appendix. While I still have a few gripes, namely,
> > > >
> > > > i) the lack of a comparison to standard adversarial training with the C&W attacks considered, to control for the confounder that the performance rests on a feature of the training removed from the novel contribution,
> > > > ii) results indicate a reduction in clean accuracy relative to baselines.
> > > > iii) evidence was not provided that results were robust to random seed
> > > >
> > > > With that being said, I agree with the authors that this constitutes the white-box evaluation I initially believed to be missing. **In the revision, I hope the authors emphasize Table 7 in the main paper, to ensure this isn't missed by the reader.**
> > > >
> > > > I also agree with the authors framing regarding how both Table 6 & 7 can be considered fair evaluations, though I do admit that I find Table 6 to be by itself unconvincing, as compared to standard adversarial training baselines, only the proposed method benefitted from the attacker's lack of knowledge about preprocessing. In Table 7, when this advantage was removed, interestingly, adversarial training was now required. However, regardless, I do acknowledge that the authors demonstrate a benefit over prior work when Table 6 & 7 are taken in totality, if we assume the results are robust to random seed. In the revision, a fair discussion regarding,
> > > > a) **which defenses in evaluation benefit from the attacker's lack of knowledge regarding preprocessing** and
> > > > b) **the need for adversarial training when the attacker does have knowledge regarding the preprocessing**,
> > > > in the main paper would be beneficial to ensure the reader has appropriate context to interpret the empirical benefit of the proposed contribution.
> > > >
> > > > Thanks for clarifying that in the detection experiments in the main paper, the attack was able to pass gradients through $R(x')$ where $x'$ is the adversarial. **I stand by the claim that this was unclear in the submission, so please ensure that this is emphasized in the revision.**
> > > >
> > > > I also agree that while an adaptive attack which is given access to the KDE would be interesting, it isn't necessary, given the comparison is still fair regardless.
> > > >
> > > > **I applaud the authors for still considering adaptive attacks in the detection scenario, and would suggest that said results are highlighted in the revision.**
> > > >
> > > > In all, I thank the authors for clarifying my misconceptions, and I hope they can improve the clarity in the revision such that future readers do not have similar misconceptions. **With that clarified, I am willing to raise my score.**

---

> > > > > ### Author Response · Authors · 2021-08-26
> > > > > **Response to Reviewer mKY1:**
> > > > >
> > > > > Thanks for your quick response and support to our work! We will keep improving the clarity as you suggested to avoid misconceptions in the revision and include all the new results. Here are our replies to your new concerns.
> > > > >
> > > > > **(1) the lack of a comparison to standard adversarial training with the CW attacks considered, to control for the confounder that the performance rests on a feature of the training removed from the novel contribution**
> > > > >
> > > > > In the table below, we report the comparison to adversarial training with the C\&W attacks under the white box setting, as an extension of Table 7.
> > > > >
> > > > > |Methods|Clean|Unseen (Mean)|$\ell _ \infty$|$\ell _  2$|JPEG|ReColor|StADV |
> > > > > |:--|:--|:--|:--|:--|:--|:--|:--|
> > > > > |AT PGD-$\ell _ \infty $ |86.8|27.2| **49.0**| 19.2|30.2|54.5| 4.8|
> > > > > |AT CW-$\ell _ \infty $ |**87.1**|23.8|43.7|17.7|28.1|45.8|3.7|
> > > > > |Ours-$\ell _ \infty $| 81.2 |**51.4**| 40.5| **43.1** | **62.1** | **73.1** | **27.4** |
> > > > >
> > > > > The results show that adversarial training (AT) with C\&W attack achieves similar performance to that with PGD attack. Our method still achieves the highest defense accuracy over unseen attacks. So the performance improvement of our method does not come from the C\&W attack.
> > > > >
> > > > > **(2) results indicate a reduction in clean accuracy relative to baselines.**
> > > > >
> > > > > The reduction in clean data accuracy is caused by removing the class-essential information in $R(x)$ because the adversarial attacks mainly distort this information, as discovered in Sec. 3.2. So the trade-off between clean data accuracy and robustness to adversarial attacks is inevitable. That being said, one can tune the trade-off by changing $\gamma$ in Eq. (8), which is another advantage of our method. By decreasing $\gamma$, $G(x')$ will retain more class-relevant information, so the clean data accuracy will be improved and the robustness will degrade. We presented an analysis of the trade-off in Sec. 4.1.
> > > > >
> > > > > **(3) evidence was not provided that results were robust to random seed**
> > > > >
> > > > > We evaluate the performance under different random seeds for the model initialization and the random sampling of VAE's bottleneck embedding. The table below reports the **error bars** (mean$\pm$std over five random trials) of AUC score for our detection method in Table 4:
> > > > >
> > > > > Method|FGSM|BIM|C\&W|PGD-$\ell _ \infty$|PGD-$\ell _ 2$
> > > > > ------|----|---|---|---|---
> > > > > KD ($R(x)$)|89.69$\pm$1.55   |99.27$\pm$0.10  |98.73$\pm$0.22  |99.30$\pm$0.09|99.32$\pm$0.11
> > > > > LID ($R(x)$)|98.59$\pm$0.59 |97.29$\pm$0.94|95.10$\pm$0.91|97.58$\pm$0.98|97.38$\pm$0.90|
> > > > > MD ($R(x)$)|99.36$\pm$0.16|99.74$\pm$0.03|99.68$\pm$0.04 |99.79$\pm$0.03|99.77$\pm$0.04|
> > > > >
> > > > > It shows that the variance caused by the randomness is small, especially for MD ($R(x)$), which achieves <0.2\% standard deviation.
> > > > >
> > > > > **(4) In the revision, a fair discussion regarding, a) which defenses in evaluation benefit from the attacker's lack of knowledge regarding preprocessing and b) the need for adversarial training when the attacker does have knowledge regarding the preprocessing in the main paper would be beneficial to ensure the reader has appropriate context to interpret the empirical benefit of the proposed contribution.**
> > > > >
> > > > > Thanks for the helpful advice! We will add corresponding discussion regrading these two points in the revision.
> > > > >
> > > > > **(5)I hope they can improve the clarity in the revision such that future readers do not have similar misconceptions.**
> > > > >
> > > > > We will improve the clarity in the revision to avoid similar misconceptions.

---

> > > > > > ### Comment · Reviewer_mKY1 · 2021-08-28
> > > > > > **Response**
> > > > > >
> > > > > > Thanks for responding to my additional concerns. I'd suggest for the revision, the significant amount of content in the author responses are included, as I feel all of it would make the paper clearer upon initial read, and ameliorate concerns of the reader.
> > > > > >
> > > > > > As there is nothing further on my end, I am willing to raise my score.

---

> > > > > > > ### Author Response · Authors · 2021-08-29
> > > > > > > **Thanks a lot for your support!**
> > > > > > >
> > > > > > > We are glad to hear that all your concerns have been successfully addressed in the discussion. We will add the content of our response to the revision. Thank you very much for increasing your score twice and your support is very important to us. We greatly appreciate that!

---

### Official Review · Reviewer_qF3G · 2021-07-16

**Rating:** 6
**Confidence:** 4

**Summary:**

The paper studies adversarial robustness in Deep Neural Networks. The main claim is that adversarial attacks are focusing on relatively small / sparse regions of the image which carry the essential class information. This small region can be efficiently estimated using a novel class disentanglement method based on VAEs. The paper conducts an empirical study on how attacks behaves on the class essential part R(x) and the remaining part G(x) and derive new methods both for adversarial defense and detection.

**Ethical Concerns:**

Nothing to report

**Limitations And Societal Impact:**

Yes, nothing to report for this section.

**Main Review:**

Overall I like the idea of the paper. It is well organized and readable. The idea of using disentanglement methods to address adversarial robustness is new to me. The empirical study is conducted carefully and the list of used baselines is complete (especially for the detection part).

A few concerns / questions that I have:
* The idea of the decomposition is nice. It would be interesting to see if the inductive bias induced by the competition with the classifier brings enough stability to this decomposition. Claiming that R(x) corresponds to the regions targeted by the attacker implies somehow that the decomposition is "unique".
* By design, adversarial attacks focuses on the pixels that are important for classification so it is hard to assess how good is the proposed decomposition. Could it be compared with simpler decompositions for instance based on saliency maps / grad-cam ?
* Some visualization are not that informative (delta G and delta R in Figure 2, the magnitude of the change is enough)
* The benchmarks are conducted against the major white-box attacks but it could include additional black-box attacks, especially for the detection part (+ using both BIM and PGD is a bit redundant). Query-based attacks can be a good solution for this.
* In section 3.2, the value of epsilon is not clearly stated (0.14 ?). Why did you choose this value and did you try different ones ? More generally, it would have been interesting to plot evolutions of metrics wrt epsilon for PGD / BIM / FGSM (in order to see if larger perturbation impacts more or less G(x) and study how the distortion evolves).
* The Appendix focuses on the important case where the attacker is aware of the defense / detection. IMO this part should belong to main paper and not to a supplementary part. A solution would be to summarize the contribution and move it to a section in the main part.

**Time Spent Reviewing:**

6h

---

> ### Author Response · Authors · 2021-08-10
> **Response to Reviewer qF3G: Part I**
>
> We appreciate your time and suggestions! Here are our detailed replies to your questions. We have added several experiments including comparison with saliency maps, performance under block-box attack and metrics wrt larger epsilons. All concerns have been duly addressed and so we humbly expect you can reconsider the decision.
>
> **(1) The idea of the decomposition is nice. It would be interesting to see if the inductive bias induced by the competition with the classifier brings enough stability to this decomposition. Claiming that R(x) corresponds to the regions targeted by the attacker implies somehow that the decomposition is "unique".**
>
> The objective in Eq. (1) is simple to optimize and the optimization converges fast after a few epochs. Thanks to the mutual information-bottleneck constraints between the VAE and the classifier, the VAE objective enforces the classifier to only pay attention to the most important information for classification, while the classifier's objective enforces the VAE to only reconstruct the class-redundant part. This helps to speed up the training of both models. Empirically, we do observe a **fast and stable convergence** on the VAE objective $L _ G$, the classifier objective $L _ D$, the VAE reconstruction accuracy, and the classifier's training accuracy, within only $60$ epochs (both models are trained from scratch on CIFAR10), i.e.,
>
> |Epoch:|10|20|30|40|50|60|70|
> |:------|:--|:--|:--|:--|:--|:--|:--|
> |$L_G$| 1.12 | 0.96|0.87|0.82|0.77|0.75|0.76|
> |$L_D$|  0.93 |0.66 |0.53|0.46|0.40|0.38|0.39|
> |$G(x)$ Reconstruction Accuracy (\%)|90.33|92.0|92.7|93.2|93.8|93.9|93.8|
> |$R(x)$ Classification Accuracy (\%)|67.27|76.7|81.3|83.3|85.9|86.5|85.9|
>
> **(2) By design, adversarial attacks focuses on the pixels that are important for classification so it is hard to assess how good is the proposed decomposition. Could it be compared with simpler decompositions for instance based on saliency maps / grad-cam?**
>
> Saliency map methods like grad-cam compute heatmaps from the gradients of a pre-trained model to highlight the important areas of an input image for making the output prediction and is usually used for interpretation, while our method conducts input space class-disentanglement and uses a VAE to decompose an input image into the sum of a class-essential part and a class-redundant part. Our optimization objective is specifically designed to extract these two types of information in the input space, so they are more accurate than the saliency map methods in our targeted applications.
>
> To compare CD-VAE with these methods, we apply a saliency map method [1] to an adversarial example $x'$ and use its result for defense: The pixels whose saliency values are greater than a threshold $T$ are considered as class-essential pixels while the rest pixels are class-redundant. Similar to our method in Sec 4.2, we use the class-redundant pixels for defense, which compose an image with class-essential pixels masked by zeros. The only hyper-parameter here is $T$ and we tried $T=0.01,0.05,0.1,0.2$. The defense accuracy on CIFAR-10 are reported in the table below:
>
> |Methods|Clean|PGD-$\ell _ \infty$|PGD-$\ell _  2$|C\&W-$\ell _ \infty$|C\&W-$\ell _  2$|StADV |
> |:--|:--|:--|:--|:--|:--|:--|
> |Normal |96.01|0.00|0.00|0.00|0.00|0.00|
> |Saliency Map ($T=0.01$)|19.20|13.54|13.68|13.33|12.54|13.97|
> |Saliency Map ($T=0.05$)|52.35|15.12|13.41|15.19|12.98|11.21|
> |Saliency Map ($T=0.1$)|71.39|10.29|8.37|11.00|8.12|8.59|
> |Saliency Map ($T=0.2$)|86.33|1.81|1.34|2.04|1.35|1.04|
> |Ours  |86.81| 77.05| 78.02| 77.04 |78.29 |19.41|
>
> Although Saliency Map($T=0.05$) achieves the best robustness among all Saliency Map methods, it still performs much poorer than our class-disentanglement method.
>
> **(3) Some visualization are not that informative (delta G and delta R in Figure 2, the magnitude of the change is enough)**
>
> The visualization of $\delta _ G$ and $\delta _ R$ shows the specific content in $\delta _ R$ and $\delta _ G$ which cannot be illustrated by magnitude: $\delta_ R$ shares similar texture and pattern with $\delta$ while  $\delta _ G$ looks like random noise. This can help to verify that $\delta$ mainly lies in $R(x)$ and rarely distort $G(x)$.
>
> **(4) The benchmarks are conducted against the major white-box attacks but it could include additional black-box attacks, especially for the detection part (+ using both BIM and PGD is a bit redundant). Query-based attacks can be a good solution for this.**
>
> (i) For the detection part, we conduct another experiment comparing the baselines and our method's defense against Square attack[2], which is a query-based black-box attack. The model architectures are illustrated in Sec 4.2. The attacker does not have access to model parameters or gradients but can query the output of the model. We set the $\ell _ \infty$ bound to be $8/255$ and the number of queries to be 5000. The table below shows the TNR and AUC scores of baselines and our method on CIFAR-10:
>
> |Method|TNR|AUC|
> |:------|:--|:--|
> |KD| 5.21| 51.27|
> |KD ($R(x)$)|  8.04 |59.20 |
> |LID| 5.17 | 51.75|
> |LID ($R(x)$)| 40.98 | 80.87|
> |MD| 10.44 | 63.99|
> |MD ($R(x)$)|  40.44 |82.44 |
> Our method significantly outperforms others in this tough setting: LID and MD on the original data $x$ can only achieve 51.75\% and 63.99\% AUC scores. However, by replacing $x$ with $R(x)$ generated by our method, LID and MD can achieve $>80\%$ AUC scores.
>
> (ii) For the defense part, in Table 7, AutoAttack[3] is an ensemble of four diverse attacks, including a query-based attack, i.e., Square attack[2]. In the table below, we compare our method with other baselines solely against Square attack[2] as an extension of Table 7. The models of baselines and our method are the same as illustrated in Sec. A.2. We set the $\ell _ \infty$ bound of Square attack to be $8/255$ and the number of queries to be 5000. We report the test accuracy against Square attack in the table below:
>
> |Methods|Square Attack|
> |:--|:--|
> |Normal| 0.1 |
> |AT-$\ell _ 2 $| 51.3 |
> |AT-$\ell _ \infty $| 57.9 |
> |HGD| 0.0 |
> |APE-GAN| 0.4 |
> |Ours-$\ell _ \infty $| 75.1|
>
> where we can clearly see that our method achieves accuracy 75.1\% accuracy, outperforming other method by a large margin. The second best method, AT-$\ell _ 2$ only achieves 57.9\% accuracy and other two VAE-based method i.e., HGD and APE-GAN fails when defensing Square Attack, achieving nearly \%0 accuracy.
>
> **(5) In section 3.2, the value of epsilon is not clearly stated (0.14 ?). Why did you choose this value and did you try different ones ? More generally, it would have been interesting to plot evolutions of metrics wrt epsilon for PGD / BIM / FGSM (in order to see if larger perturbation impacts more or less G(x) and study how the distortion evolves).**
>
> (i) We follow most adversarial attack papers [4-7], which set epsilon to $8/255$ for image pixels normalized to $[0,1]$. They then normalize the images with mean=(0.485, 0.456, 0.406) and standard deviation std=(0.229, 0.224, 0.225), which are the mean and std of all ImageNet images. The minimum std over the three channels is 0.224 and $l_ \infty$-norm only takes the maximum element of the whole vector, so the $l_ \infty$-ball's epsilon of adversarial perturbation accordingly changes to 0.14 ($0.14 \approx8\div 255 \div 0.224$).
>
> (ii) We try different values of epsilon (8/255, 48/255, 96/255, 144/255, 255/255) in PGD and report the $\ell _ p$-norm of $\delta$, $\delta_ G$ and $\delta _R $ in the table below:
>
> |epsilon||$\ell _ 1 \times 10 ^ {-3}$|$\ell _2$ | $\ell _ \infty$|
> |:--|:--|:--|:--|:--|
> |8/255|$\delta$| 13.65$\pm$0.88 |39.09$\pm$1.40|0.14$\pm$0.00|
> |8/255|$\delta _ G$|4.15$\pm$0.65|16.56$\pm$2.47|0.39$\pm$0.17|
> |8/255|$\delta _ R$|13.78$\pm$0.92|40.97$\pm$1.87|0.48$\pm$0.16|
> ||||||
> |48/255|$\delta$|70.10$\pm$4.04 |207.24$\pm$8.51|0.84$\pm$0.00|
> |48/255|$\delta _ G$|19.67$\pm$3.20|63.87$\pm$9.82|0.90$\pm$0.43|
> |48/255|$\delta _ R$|64.50$\pm$4.30|192.35$\pm$10.47|1.51$\pm$0.42|
> ||||||
> |96/255|$\delta$|125.91$\pm$7.41|381.01$\pm$15.66|1.68$\pm$0.00|
> |96/255|$\delta _ G$|49.67$\pm$9.73|155.67$\pm$25.50|2.74$\pm$0.32|
> |96/255|$\delta _ R$|111.42$\pm$9.05|337.48$\pm$23.83|1.61$\pm$0.30|
> ||||||
> |144/255|$\delta$| 173.23$\pm$10.86 |533.11$\pm$23.99|2.52$\pm$0.00|
> |144/255|$\delta _ G$|88.40$\pm$15.69|270.85$\pm$38.58|2.36$\pm$0.20|
> |144/255|$\delta _ R$|146.88$\pm$13.17|446.07$\pm$34.35|3.77$\pm$0.30|
> ||||||
> |255/255|$\delta$|256.76$\pm$28.93|790.63$\pm$83.14|4.43$\pm$0.07|
> |255/255|$\delta _ G$|141.29$\pm$33.16|430.33$\pm$84.50|3.30$\pm$0.25|
> |255/255|$\delta _ R$|226.98$\pm$39.10|653.33$\pm$87.96|4.70$\pm$0.47|
>
> It shows that by increasing epsilon, $G(x)$ suffers more distortion from the adversarial attacks. When epsilon is small (8/255 and 48/255), the $\ell _1$ and $\ell _2$ norm of $\delta _ G$ is around $1/3$ of that of $\delta$. When epsilon becomes large (144/255 and 255/255), the $\ell _1$ and $\ell _2$ norm of $\delta _ G$ grows, exceeding $1/2$ of that of  $\delta$. This demonstrates that with large epsilon, the attackers first distort the class-essential information in $R(x)$ and then seek to perturb class-redundant information in $G(x)$. Note epsilon larger than 48/255 is rarely used for producing adversarial examples since it drastically changes the original images. So $G(x)$ is sufficiently robust to attacks using reasonable epsilon values.

---

> > ### Author Response · Authors · 2021-08-10
> > **Response to Reviewer qF3G: Part II**
> >
> > **(6) The Appendix focuses on the important case where the attacker is aware of the defense / detection. IMO this part should belong to main paper and not to a supplementary part. A solution would be to summarize the contribution and move it to a section in the main part.**
> >
> > We agree and will move the part to the main paper.
> >
> > ***
> >
> > [1]https://github.com/Ema93sh/pytorch-saliency
> >
> > [2] Maksym Andriushchenko, Francesco Croce, Nicolas Flammarion, Matthias Hein. Square Attack: a query-efficient black-box adversarial attack via random search. ECCV, 2020.
> >
> > [3]https://github.com/fra31/auto-attack
> >
> > [4] Leslie Rice, Eric Wong, J. Zico Kolter. Overfitting in adversarially robust deep learning. ICML, 2020.
> >
> > [5] Lang Huang, Chao Zhang, Hongyang Zhang. Self-Adaptive Training: beyond Empirical Risk Minimization. ICML 2020.
> >
> > [6] Cassidy Laidlaw, Sahil Singla, Soheil Feizi. PERCEPTUAL ADVERSARIAL ROBUSTNESS: DEFENSE AGAINST UNSEEN THREAT MODELS. ICLR 2021
> >
> > [7] Hongyang Zhang, Yaodong Yu, Jiantao Jiao, Eric P. Xing, Laurent El Ghaoui, Michael I. Jordan. Theoretically Principled Trade-off between Robustness and Accuracy. ICML, 2019.

---

> > > ### Comment · Reviewer_qF3G · 2021-09-01
> > > **Response**
> > >
> > > Thanks a lot for answering my questions! Indeed for 2), there's not a lot of decomposition in input space to compare with but IMO, it's still interesting to see the gap with a "naive" method such as the saliency map. Thanks for all the additional experiments, especially the dynamic wrt to epsilon and the separated experiments with black-box attacks.
> > >
> > > Thanks more generally for all the additional work done for this rebuttal!

---

> > > > ### Author Response · Authors · 2021-09-01
> > > > **[Last day discussion] Would you mind reconsidering your rating given that All the concerns are addressed and Five groups of new experiments are reported?**
> > > >
> > > > Dear Reviewer qF3G,
> > > >
> > > > Thanks for your feedback! Since **all your concerns are well addressed** with **Five new groups of experiments posted**, and you are positive about the new experimental results and the answers they indicate, would you mind reconsidering your original rating?
> > > >
> > > > We believe that our method is simple, effective, and principal in addressing a novel open problem, i.e., input-space task/class-disentanglement, which is of general interest to many fundamental machine learning problems. **Your support is very important to this new topic and new method during their earlier stages!** We greatly appreciate it!
> > > >
> > > > Best Regards,
> > > >
> > > > Authors

---

### Official Review · Reviewer_FeME · 2021-07-16

**Rating:** 7
**Confidence:** 4

**Summary:**

The manuscript proposed a method for the detection and defense against adversarial attacks in classifier models. Towards this goal, a method is proposed to break down inputs $x$  into two components $x=G(x) + R(x)$ where the $R(x)$ contains relevant (discriminative) class-level information and $G(x)$ contains more redundant information shared among classes.
The main observation behind the proposed methods is that adversarial attacks target task-essential information ($R(x)$ in this case) and have reduced effect in task-redundant information ($G(x)$).

The manuscript shows that adversarial attacks can be effectively detected by focusing on R(x). Likewise,  robustness towards these attacks, .i.e defenses, can be achieved feeding the classifier with $G(x)$ instead of $x$.

**Ethical Concerns:**

N.A.

**Limitations And Societal Impact:**

N.A.

**Main Review:**

The content of the manuscript is clear and it has a good flow.
The proposed method is sound, interesting and the idea of dividing the input space into the relevant ($R(x)$) and redundant ($G(x)$) components is, to the best of my knowledge, novel.

I find the main strengths of the proposed method to lie on its simplicity and its characteristic of being attack-agnostic (no need to have prior knowledge of possible attack methods).

My main concerns with the manuscript are the following:

-C1: The idea of adversarial attacks targeting specific class-relevant information is not that novel. While the exploitation of this idea is different, to a good extent the proposed method has similarities with Granda et al., arxiv:2010.15974. Positioning the proposed method with respect to that work would strengthen the manuscript and further highlight the novel aspects of the proposed method.

-C2: in L.72 it is stated that through the proposed method, of focusing on $R(x)$, it is possible to analyze the behavior of neural networks and shed novel insights that cannot be revealed by latent factors. I find this statement not properly supported by experiments in the evaluation section. Moreover, while this can be considered a difference between the input space and latent representations, through the use of visualization algorithms it is now possible to check visually these latent representations.
In my opinion the statement in L.72 should be toned down.

-C3: In L.100 it is stated that models that train a binary classification networks (detector) on top of the internal representation of the model to be defended may fail if the detector model is leaked. Doesn't the proposed method also suffer from the same problem? what if the used VAE used to extract the component $G(x)$ is leaked?
Perhaps this part of the text needs some rephrasing, in its current form it gives the impression that this is a weakness of these methods but not of the proposed one.


-C4: Perhaps I missed something, it seems that the defense/robustness against adversarial attacks (Sec. 3.4) is to use as input $G(x)$ instead of $x$. However, if critical/discriminative class-related information is not in $G(x)$ (since it is in $R(x)$) wouldn't it be sacrificing, significantly, the performance of the model (classifier) to be defended. This is already suggested by the results in Table 3 where the model is trained with $x$ and tested with $G(x)$. Since this seem to be the case, the question is whether this drop in predictive performance is a good trade-off to pay for the sake of robustness.

-C5: When evaluating the proposed adversarial defense (Sec.4.3) a new set of spatial transform-based attacks are considered which were not considered for the evaluation of the detection method (Sec. 4.2). I find this a bit counter-intuitive. Is there a good motivation for this? Having a consistent set of baselines and settings will strengthen the validation of the proposed method.

==================
Post Rebuttal Update
==================

I thank authors for addressing my feedback on their rebuttal.
I went through other reviews and the provided rebuttal.

Overall the provided response addressed most of my criticisms to a good extent.

As part of point (1) the rebuttal claims that the proposed method can also be used for interpretation of DNNs. I find content supporting this claim to be very weak. Likewise, the second point (2) of the response hints at this interpretation capability can be achieved through the inspection of $R(x)$. Looking at Fig.2 presented in the paper, I find the visualizations of $R(x)$ somewhat ambiguous. I struggle to see how $R(x)$ can serve as means of analyzing the behavior of neural networks. I am still of the thought that the interpretation capabilities of the proposed method (claimed in the paper) are somewhat unjustified.
In my opinion the claims made by the manuscript along the interpretation aspects should be be toned down.

Keeping the comments from above I have decided increasing my initial rating.


**Time Spent Reviewing:**

3

---

> ### Author Response · Authors · 2021-08-10
> **Response to Reviewer FeME:**
>
> We appreciate your time and suggestions! Here are our detailed replies to your questions. All concerns have been duly addressed and so we humbly expect you can reconsider the decision.
>
> **(1) The idea of adversarial attacks targeting specific class-relevant information is not that novel. While the exploitation of this idea is different, to a good extent the proposed method has similarities with Granda et al., arxiv:2010.15974.**
>
> Our method is very different from Granda et al.[1] in several fundamental aspects:
>
> (i) Granda et al.[1] study the class-relevant filters/latent features that adversarial attacks mainly target, while we study the class-relevant pixels in the **input space** because the adversarial attacks are optimized to perturb the input pixels. Our input-space class-disentanglement also provides more straightforward interpretations to the classifier and attacks.
>
> (ii) Granda et al.[1] need to solve a constrained optimization problem to select filters and detects adversarial examples based on these selected filters, which is empirically difficult and expensive. On the contrary, our method performs class-disentanglement by solving a **simple unconstrained optimization** in Eq. (1), which is easier and faster. Our mutual information-bottleneck constraints between the VAE and the classifier further speed up the training.
>
> (iii) Granda et al.[1] only address the adversarial detection problem. Instead, our method can be utilized for a **broad class of tasks**, e.g., adversarial detection, adversarial defense, interpretation of DNN classifier and adversarial attacks, etc. Moreover, it is complementary to and can be easily incorporated with existing methods of these tasks, e.g., by replacing their input $x$ with $G(x)$ or $R(x)$.
>
> **(2) In L.72 it is stated that through the proposed method, of focusing on $R(x)$, it is possible to analyze the behavior of neural networks and shed novel insights that cannot be revealed by latent factors. I find this statement not properly supported by experiments in the evaluation section. Moreover, while this can be considered a difference between the input space and latent representations, through the use of visualization algorithms it is now possible to check visually these latent representations.**
>
> The statement in L.72 intends to claim that the input-space class-disentanglement can reveal new properties. We will modify the statement to be more precise.
>
> For example, adversarial attacks perturb the input pixels. So we decompose the input pixels into the sum of two images, where one ($R(x)$) is mainly attacked and the other ($G(x)$) is not. But it is unknown how to generate two visualization heatmaps whose sum recovers the original image. The heatmaps can neither be used to train classifiers as in Table 3 for empirical analysis in Sec. 3.2, which reveals the key properties inspiring our adversarial detection and defense strategy.
>
> In addition, the visualization result associates with the latent factors of a specific classifier and an attack: so we need to regenerate it if any of them change. On the contrary, the properties of input-space class-disentanglement in Sec. 3.2 can be generally verified on different classifiers and attacks without retraining our CD-VAE model because our disentanglement is not generated for a specific (attacked) model and a specific attack. This also explains why our detection and defense methods consistently perform well over all the evaluated attacks in Table 5-7, while other baselines can only work for some attacks but fail on some others.
>
> **(3) In L.100 it is stated that models that train a binary classification networks (detector) on top of the internal representation of the model to be defended may fail if the detector model is leaked. Doesn't the proposed method also suffer from the same problem? what if the used VAE used to extract the component $G(x)$ is leaked?**
>
> We will rephrase this statement. It is well known that the detection performance degrades when the detection model is leaked. Since our strategy applies an existing detection method to $R(x)$, we cannot avoid such problems inherited from most existing detection methods.
>
> **(4) It seems that the defense/robustness against adversarial attacks (Sec. 3.4) is to use as input $G(x)$ instead of x. However, if critical/discriminative class-related information is not in $G(x)$  (since it is in $R(x)$) wouldn't it be sacrificing, significantly, the performance of the model (classifier) to be defended. Since this seem to be the case, the question is whether this drop in predictive performance is a good trade-off to pay for the sake of robustness.**
>
> We agree that the trade-off between the robustness and the clean-data accuracy always exists for any adversarial defense method, as shown in Table 3. To avoid a significant drop in the clean-data accuracy, we can control this trade-off by adjusting $\gamma$ in Eq. (1), which is a trade-off hyperparameter between VAE reconstruction and classification. We presented an empirical analysis of this trade-off in Sec. 4.1 and Fig. 4, which show that a sweet spot for a good trade-off does exist. In Table 6, we tune $\gamma$ on a validation set for each dataset and choose $\gamma=1e-2$ for CIFAR-10 and $\gamma=1$ for restricted ImageNet. This results in a clean data accuracy of 86.8\% on CIFAR-10 and 65.3\% on restricted ImageNet, which are comparable to the baselines. Meanwhile, we achieve much better robustness than the baselines, e.g., >70\% defense accuracy on CIFAR-10 and >50\% on restricted ImageNet against $\ell _ p$ norm bounded attack.
>
> **(5) When evaluating the proposed adversarial defense (Sec.4.3) a new set of spatial transform-based attacks are considered which were not considered for the evaluation of the detection method (Sec. 4.2). Is there a good motivation for this? Having a consistent set of baselines and settings will strengthen the validation of the proposed method.**
>
> The attacks for evaluating adversarial defense and adversarial detection are different because we follow the benchmark settings adopted by most works from these two different areas, e.g., the state-of-the-art methods for adversarial detection[2-5] and adversarial defense[6,7], respectively. Since we compare with the results reported in their papers, for fair comparisons, we did not modify the two areas' settings to be the same.
>
> ***
>
> [1] Roger Granda, Tinne Tuytelaars, Jose Oramas. Can the state of relevant neurons in a deep neural networks serve as indicators for detecting adversarial attacks? arxiv.org:2010.15974
>
> [2] Jinyu Tian, Jiantao Zhou, Yuanman Li,Jia Duan. Detecting Adversarial Examples from Sensitivity Inconsistency of Spatial-Transform Domain. AAAI, 2021.
>
> [3] Zhijie Deng, Xiao Yang, Shizhen Xu, Hang Su1, Jun Zhu. LiBRe: A Practical Bayesian Approach to Adversarial Detection. CVPR, 2021.
>
> [4] Kimin Lee, Kibok Lee, Honglak Lee, Jinwoo Shin. A Simple Unified Framework for Detecting Out-of-Distribution Samples and Adversarial Attacks. NeurIPS, 2018.
>
> [5] Xingjun Ma, Bo Li, Yisen Wang, Sarah M. Erfani, Sudanthi Wijewickrema, Grant Schoenebeck, Dawn Song, Michael E. Houle, James Bailey. CHARACTERIZING ADVERSARIAL SUBSPACES USING LOCAL INTRINSIC DIMENSIONALITY. ICLR, 2018.
>
> [6] Cassidy Laidlaw, Sahil Singla, Soheil Feizi. PERCEPTUAL ADVERSARIAL ROBUSTNESS: DEFENSE AGAINST UNSEEN THREAT MODELS. ICLR 2021.
>
> [7] Dawei Zhou, Tongliang Liu, Bo Han, Nannan Wang, Chunlei Peng, Xinbo Gao. Towards Defending against Adversarial Examples via Attack-Invariant Features. ICML 2021.

---

> > ### Comment · Reviewer_FeME · 2021-09-01
> > **Final impression**
> >
> > I thank authors for addressing my feedback on their rebuttal.
> > I went through other reviews and the provided rebuttal.
> >
> > Overall the provided response addressed most of my criticisms to a good extent.
> >
> > As part of point (1) the rebuttal claims that the proposed method can also be used for interpretation of DNNs. I find content supporting this claim to be very weak. Likewise, the second point (2) of the response hints at this interpretation capability can be achieved through the inspection of $R(x)$. Looking at Fig.2 presented in the paper, I find the visualizations of $R(x)$ somewhat ambiguous. I struggle to see how $R(x)$ can serve as means of analyzing the behavior of neural networks. I am still of the thought that the interpretation capabilities of the proposed method (claimed in the paper) are somewhat unjustified.
> > In my opinion the claims made by the manuscript along the interpretation aspects should be be toned down.
> >
> > Keeping the comments from above I have decided increasing my initial rating.

---

> > > ### Author Response · Authors · 2021-09-01
> > > **Thanks a lot for your support!**
> > >
> > > We are glad to hear that most of your criticisms are addressed to a good extent. We will keep improving the accuracy of our claims as you suggested. We will add detailed textual explanations to the visualization results and adjust the claims about the interpretation of $R(x)$. In short, in the visualizations, $R(x)$ only preserves non-robust, high-frequency, and sparse patterns like contours or corners/edges as essential features of the class, but it usually excludes the backgrounds, light conditions, and some colors.
> > >
> > > Thank you very much for increasing your score and your support is very important to us. We greatly appreciate that!

---

> ### Author Response · Authors · 2021-08-30
> **New experiments on White-box detection show our advantage when G is leaked (Answering C3 in your comments). Would you mind checking it and reconsidering the rating?**
>
> Dear Reviewer FeME,
>
> Thanks for the previous comments! We have reported a **new experiment on white-box detection**, i.e., when both the classifier and the detector models (including VAE $G(\cdot)$ in our method) are leaked. **This directly addresses C3 in your concern. It shows that our method can bring even more significant improvement to the detection accuracy in this challenging setting**. We believe that all your concerns have been duly addressed with new experiments, so we humbly expect you can reconsider the rating. Here are the details of the new experiments.
>
> **Our detection strategy can bring significant improvements under adaptive/white-box attacks**, which target both the classifier and the detector. This can be seen in the table below: in the white-box setting, our method using $R(x)$ as input significantly outperforms the baseline using $x$ as input. In particular, we apply our strategy to MD[1] because MD[1] considers adaptive attacks in their paper. We follow their setting, i.e., Sec. E of [1], and use PGD attack to generate adversarial images, which maximizes the classification loss and minimizes the Mahalanobis distance at the same time. The detection accuracy on CIFAR-10 is reported in the table below and the features being attacked concatenate the outputs of all the residual blocks in WideResNet.
>
> ||$\ell _ \infty$||$\ell _2$||
> |:------|:--|:--|:--|:--|
> || TNR | AUC|TNR|AUC|
> |MD|  27.64 |77.63 |27.16|75.15|
> |MD($R(x)$) |**59.25**|**90.66**|**87.97**|**97.72**|
>
> Thanks a lot for your feedback! Any further discussion/question is welcomed!
>
> Best Regards,
>
> Authors
>
> ***
>
> [1] Kimin Lee, Kibok Lee, Honglak Lee, Jinwoo Shin. A Simple Unified Framework for Detecting Out-of-Distribution Samples and Adversarial Attacks. NeurIPS, 2018

---

### Official Review · Reviewer_kR37 · 2021-07-17

**Rating:** 6
**Confidence:** 4

**Summary:**

The paper proposes an approach to separate class-specific and irrelevant information from an input image. A VAE is used to reconstruct irrelevant information. The VAE is jointly trained with a classifier to achieve this separation. The paper concludes that the class-specific signal can be used for adversarial detection while the irrelevant information can be used for adversarial defense.

**Ethical Concerns:**

None.



**Limitations And Societal Impact:**

Limitations and societal impact are not discussed. These considerations are very much relevant to the nature of the work.

**Main Review:**

Strengths:
- This is an input processing defense approach where a VAE is first applied to reconstruct an input image that is sufficient for classification. Subsequently a classifier operated on the non-redundant information and classifies the reconstructed image.
- Information bottleneck prinicpal is applied to achieve disentanglement in the input space.
- An interesting relationship with the adversarial examples is shown i.e., the perturbations mostly relate to the information unrrelated to the image class.

Weaknesses:
- The comparisons with other detection and defense mechanisms mostly include old approaches (pre 2018) and do not compare with the state of the art defense mechanisms.
- The main contribution of this work is to apply Information Bottleneck principal on the latent representation of a VAE. There exists previous VAE based approaches for adversarial defense, as I elaborate in a later comment. Therefore, the novelty is limited to the combination of beta-VAE with the IB loss.
- Section A.2 in supp. material mentions the values of hyper-parameters used for training, how are these values set?
- Paper is written in an obscured way at several important points. E.g., the abstract can be improved for clarity and by avoiding notations that are not required as such at the very beginning.
- AutoAttack and other white box attack results are provided in the supplementary material. I believe these are the main results and should be included in the main paper.
- The paper does not discuss and compare with relevant literature. For example, there exist a number of approaches based on VAE as an input processing step [A,B]. The proposed technique should be discussed in the context of these works and compared with the existing related methods.
- Do the authors plan to release the code? There are several implementation details that are less clear and a codebase will help reproduce the reported results.

[A] Defense-VAE: A Fast and Accurate Defense against Adversarial Attacks
[B] PuVAE: A Variational Autoencoder to Purify Adversarial Examples

Post-rebuttal Comments: The authors have done a great job of answering the raised queries. I am therefore inclined towards accepting this paper and request authors to make the promised changes.

**Time Spent Reviewing:**

8

---

> ### Author Response · Authors · 2021-08-10
> **Response to Reviewer kR37:**
>
> We appreciate your time and comments! We have compared our method with the latest and state-of-the-art detection and defense mechanisms as you suggested. Here are our detailed replies to your questions. All concerns have been duly addressed and so we humbly expect you can reconsider the decision.
>
> **(1) The comparisons with other detection and defense mechanisms mostly include old approaches (pre 2018) and do not compare with the state-of-the-art defense mechanisms.**
>
> (i) In the table below, we compare our method with SID[1] (published in 2021), which is the latest detection method we could find. We evaluate our detection of three types of attacks on CIFAR-10 and compare the detection accuracy with the results reported in SID[1]. Our method outperforms SID[1] by a large margin across all these attacks. Moreover, our method is complementary to most detection methods and can generally improve them by simply replacing $x$ with $R(x)$ in their methods.
>
> |Methods|FGSM|BIM|C\&W|
> |:--|:--|:--|:--|
> |SID[1] (AAAI 2021)| 94.79 | 99.38 |96.56|
> |MD-$R(x)$  (Ours)|  **99.36** | **99.74** | **99.68** |
>
> (ii) In the table below, we compare our method with three latest and state-of-the-art defense methods [2,3,4] on the more challenging white-box attacks, as an extension of the experiments in A.2 of the Appendix. We download the best checkpoints of their models trained with $l _ \infty$ attacks (bounded by 8/255) and compare their defense accuracy with ours. Our method achieves the highest "Unseen (Mean)" (the defense accuracy averaged over all the attacks NOT used for adversarial training of the defense model). Our method also achieves the highest defense accuracy against three out of the five attacks, i.e., $l_2$, JPEG, and ReColor.
>
> |Methods|Clean|Unseen (Mean)|$\ell _ \infty$|$\ell _  2$|JPEG|ReColor|StADV |
> |:--|:--|:--|:--|:--|:--|:--|:--|
> |SAT-$\ell _ \infty $[2] (NeurIPS 2020) |**85.6**|32.4|**53.1**|23.4|35.9|66.5|3.9|
> |OIA-$\ell _ \infty $[3] (ICML 2020) |83.4|33.6|51.6|24.2|35.4|67.8|7.0|
> |PAT-$\ell _ \infty $[4] (ICLR 2021) |82.4|50.3|30.2|34.9|48.7|71.0|**46.4**|
> |Ours-$\ell _ \infty $| 81.2 |**51.4**| 40.5| **43.1** | **62.1** | **73.1** | 27.4 |
> |Ours-$\ell _ 2 $|  81.0 | 50.4| 39.4 | 42.4 | 61.6 | 72.2 | 28.4 |
>
> **(2) Subsequently a classifier operated on the non-redundant information and classifies the reconstructed image.**
>
> The classifier in CD-VAE is not used to classify the VAE reconstructed image $G(x)$. In CD-VAE objective (Eq. (1)-(3)), it is applied to $R(x)=x-G(x)$ for extracting the class-essential information. We do not use it for any other classification purpose in the paper.
>
> **(3) The main contribution of this work is to apply Information Bottleneck principal on the latent representation of a VAE.**
>
> (i) We respectfully disagree with your justification for our main contribution. Our main contribution is to develop a class-disentanglement model in the **input space** instead of a **latent space** and the goal is to decompose an input image into a class-dependent part plus a class-independent part in the pixel space.
>
> (ii) Instead of directly applying the information bottleneck principle to VAE, we formulate class-disentanglement as a **mutual** information-bottleneck problem, i.e., the VAE and the classifier's objectives in Eq. (1)-(3) performs as information-bottleneck constraints for each other. This also makes the joint optimization simpler and faster.
>
> **(4) There exists previous VAE based approaches for adversarial defense, as I elaborate in a later comment. Therefore, the novelty is limited to the combination of beta-VAE with the IB loss. For example, there exist a number of approaches based on VAE as an input processing step [A,B].**
>
> (i) Our method is significantly different from [A,B] ([5,6] in the reference below) on several fundamental aspects, e.g., the motivations (input space class-disentanglement), the optimization formulation (Eq. (1)), and the targeted applications.
>
> (ii) For example, these VAE based methods fail on defending white-box attacks (when their VAE models are exposed to the attackers). On the contrary, as shown in Table 7 from A.2 of the Appendix, our method outperforms most baselines when defending different types of unseen white-box attacks. This difference is due to our class-disentanglement model, which is not considered in these methods.
>
> (iii) Moreover, these VAE methods cannot be used for both adversarial attack and defense: they are only developed for the defense task. Thanks to the mutual information bottlenecks, our method "kills two birds with one stone", i.e., we conduct adversarial detection on $R(x)$ and adversarial defense on $G(x)$. We will add discussions to these works.
>
> **(5) How are these values (hyperparameters) set?**
>
> There are only two hyper-parameters in CD-VAE, i.e., $\beta$ and $\gamma$, and we tune them on a small held-out validation set of CIFAR-10, which is composed of 2000 images from the original training set. Note $\beta$ is a hyperparameter from $\beta$-VAE. For the newly introduced hyperparameter $\gamma$, we conducted a sensitivity analysis in Sec. 4.1.
>
> **(6) AutoAttack and other white-box attack results are provided in the supplementary material. I believe these are the main results and should be included in the main paper.**
>
> We agree and will move the part to the main paper.
>
> **(7) Paper is written in an obscured way at several important points. E.g., the abstract can be improved for clarity and by avoiding notations that are not required as such at the very beginning.**
>
> We will follow your suggestions and polish up our writing.
>
> **(8) Do the authors plan to release the code? There are several impelementation details which are less clear and a codebase will help reproduce the reported results.**
>
> Yes, we will release our code later. We believe it will be a general tool for many ML tasks.
>
>
> ***
>
> [1] Jinyu Tian, Jiantao Zhou, Yuanman Li, Jia Duan. Detecting Adversarial Examples from Sensitivity Inconsistency of Spatial-Transform Domain. AAAI, 2021.
>
> [2] Lang Huang, Chao Zhang, Hongyang Zhang. Self-Adaptive Training: beyond Empirical Risk Minimization. ICML 2020.
>
> [3] Leslie Rice, Eric Wong, J. Zico Kolter. Overfitting in adversarially robust deep learning. ICML, 2020.
>
> [4] Cassidy Laidlaw, Sahil Singla, Soheil Feizi. PERCEPTUAL ADVERSARIAL ROBUSTNESS: DEFENSE AGAINST UNSEEN THREAT MODELS. ICLR 2021
>
> [5] Xiang Li, Shihao Ji. Defense-VAE: A Fast and Accurate Defense against Adversarial Attacks. MLCS, 2019.
>
> [6] Uiwon Hwang, Jaewoo Park, Hyemi Jang, Sungroh Yoon, Nam Ik Cho. PuVAE: A Variational Autoencoder to Purify Adversarial Examples. arXiv:1903.00585

---

> > ### Comment · Reviewer_kR37 · 2021-08-27
> > **Thanks for the responses!**
> >
> > I thank the authors for providing responses to my queries. I believe most of the comments are addressed and there are some promised changes that will hopefully improve the clarity of the approach. I am therefore improving my rating.

---

> > > ### Author Response · Authors · 2021-08-28
> > > **Thanks a lot for your support!**
> > >
> > > We are glad to hear that most of your comments have been addressed in our response. We will keep improving the clarity of the paper and modify it as you suggested. Thank you very much for increasing your score and your support is very important to us. We greatly appreciate that!

---

> ### Author Response · Authors · 2021-08-26
> **Comparisons to Four new baselines and answers to all your concerns. Would you mind checking them and reconsidering the decision?**
>
> Dear Reviewer kR37,
>
> Thanks for the comments! Here is a summary of our detailed response below. We humbly expect you can check it and reconsider the decision.
>
> * We add **new experimental results comparing our method with Four baselines published in 2020 and 2021**. Our method still consistently outperforms them by a large margin.
>
> * Our main novelty on the methodology is not simply beta-VAE + IB loss. In fact, **the novelty is (1) input space disentanglement and (2) mutual IB constraints between a beta-VAE and a classifier**.
>
> * **We only have two hyperparameters**: (1) $\beta$ is inherited from beta-VAE and (2) how to choose $\gamma$ is thoroughly analyzed in Sec. 4.1.
>
> * We will carefully polish the writing and organization of this paper according to your suggestions.
>
> * Please check our responses to other reviewers, which report a great number of new experiments further supporting our claims about the advantages of CD-VAE in different scenarios.
>
> Thanks a lot for your feedback! Any further discussion/question is welcomed! Your support for a novel, simple, and principal method in its earlier stage is very important and we sincerely appreciate it!
>
> Best Regards,
>
> Authors

---

### Official Review · Reviewer_TtXL · 2021-07-21

**Rating:** 7
**Confidence:** 3

**Summary:**

Inspired by information bottleneck, the paper proposed a framework that creates class-disentanglement in the input space by setting up a competition between VAE reconstruction on G(x) and accurate classification on x-G(x). Then empirical observations suggest that popular adversarial attack methods mainly affect the task-essential information x-G(x) and has less influence on task-redundant part G(x). Such an observation then motivates a new strategy towards adversarial detection and defense. Extensive empirical study on CIFAR-10 and Imagenet corroborate the hypothesis.

**Limitations And Societal Impact:**

No potential negative societal impact was discussed.

**Main Review:**

Different from existing work on information bottleneck over latent space, this paper presents an interesting framework trying to learn class-disentanglement on the input space. The observations that adversarial attack mainly affects the task-essential information is quite interesting and indeed leads to practical algorithms for adversarial attack detection and defense, which empirically works better based on the provided results. Below are my detailed comments/questions:

- Since this is an empirical study and few formal definitions/theorems are provided, I found some concepts/definitions are not clear. For example, in line 61, "G(x) contains all the information redundant or irrelevant to the class y".

What is redundant information? I guess the redundant information in G(x) is task-relevant but redundant/same with respect to the information in x-G(x). Then if x-G(x) is negatively affected by the adversarial attack, this part of information in G(x) should also be easily affected? Why is G(x) robust then?

If the task irrelevant information means information that is not directly related to the classification task such as background color, then adversarial attack such as PGD should not change this part too much since they do not affect the class prediction too much.

Thus G(x), as a mixture of both, seems to be affected less but the important part in G(x) may still be affected?
- One advantage of existing information bottleneck methods on latent space is that, since raw sample x may contain many task-irrelevant information and the useful information likely lies in a submanifold of original sample space, it is desirable to extract task-essential information in a low-dimensional latent space and learn a downstream classifier purely on the latent space (e.g. improve data efficiency). While in the proposed approach, the classification has to be performed on a space with the same dimension as the original data space (either x, G(x), or x-G(x)).
- The PGD attack is with respect to a pre-trained classifier on clean data x, not the classifier in CD-VAE on x-G(x), right? Because this will make sure x-G(x) is vulnerable to adversarial attack not merely because we are attacking D(x-G(x)).
- For adversarial attack restricted to lp-norm, there has been rich literatures on certified defense, i.e. defense with theoretical guarantee. Since this paper also focuses on lp-norm attack, formal theoretical discussions on the proposed method will greatly improve the paper. For example, in line 116, the paper claims "we show that classification on the class-redundant part G(x) is robust to adversarial attacks". Is this true for all kinds of attack? What is the assumptions needed? And if we can extract class-redundant part in the latent space using IB, is that also robust to adversarial attack? Any experimental results on this?
- In Eq 2, why do we need disentanglement (beta-VAE) in the feature space of VAE. Such a disentanglement notion seems irrelevant to the class disentanglement discussed in this paper.
- No error bars are provided for all the tables, which may be needed to demonstrate that the empirical findings are not from randomness (especially for a pure empirical paper).

**Time Spent Reviewing:**

3

---

> ### Author Response · Authors · 2021-08-10
> **Response to  Reviewer TtXL**
>
> We appreciate your time and suggestions! We have added an experiment of CD-VAE in latent space and provided error bars of the detection part. Here are our detailed replies to your questions.  All concerns have been duly addressed and so we humbly expect you can reconsider the decision.
>
> **(1) I found some concepts/definitions are not clear. For example, in line 61, "G(x) contains all the information redundant or irrelevant to the class y".**
>
> $G(x)$ captures the information not required by the classifier in CD-VAE, which covers the class-irrelevant information. But some class-redundant information can also be preserved in $G(x)$, as implied by the test accuracy of using $G(x)$ for training in Table 3. The amount of class-redundant information can be controlled by adjusting the reconstruction-classification trade-off weight $\gamma$ in Eq. (1). Sec. 4.1 and Fig. 4 presented a thorough sensitivity analysis of $\gamma$ for different applications. When $\gamma$ is large, $G(x)$ tends to only preserve the information irrelevant to the class. When $\gamma$ is small, some class-redundant information can be preserved in $G(x)$.
>
> **(2) What is redundant information? I guess the redundant information in G(x) is task-relevant but redundant/same with respect to the information in x-G(x). Then if x-G(x) is negatively affected by the adversarial attack, this part of information in G(x) should also be easily affected? Why is G(x) robust then?**
>
> In our empirical study of Sec. 3.2, we show that the class-redundant information in $G(x)$ is **useful for classification** but is very **different from** the information used by the classifier trained on $x$ or $R(x)$, where the latter is the focus of adversarial attacks.
>
> In particular, Table 3 shows that the test accuracy of training a classifier on $G(x)$ can still be high, indicating that $G(x)$ is class-relevant. However, Table 2 (comparing the last three rows) and Fig. 2 show that $G(x)$ is much less distorted by the adversarial attacks than $R(x)$. It implies that $G(x)$ is robust to attacks and has been demonstrated by the outstanding performance of our $G(x)$-based defense in Table 6-7.
>
> We also explained this phenomenon via Table 3 and the analysis below it. One can look at the test accuracy of a classifier trained on $G(x)$ but tested on $x$, $R(x)$, or $G(x)$: the accuracy is low if the classifier takes $x$ or $R(x)$ as input but is high when taking $G(x)$ as input. It indicates that $G(x)$ does preserve class-relevant information (because of the high test accuracy on $G(x)$) but this information is orthogonal to and very different from the class-relevant information the attacked classifiers (trained on $x$) heavily rely on, where the latter is the main focus of adversarial attacks since the attacks are optimized to degenerate the classifier trained on $x$. Since the class-relevant information in $G(x)$ is very different, it is robust to these attacks.
>
> **(3) One advantage of existing information bottleneck methods on latent space is that, since raw sample x may contain many task-irrelevant information and the useful information likely lies in a submanifold of original sample space, it is desirable to extract task-essential information in a low-dimensional latent space and learn a downstream classifier purely on the latent space (e.g. improve data efficiency). While in the proposed approach, the classification has to be performed on a space with the same dimension as the original data space (either x, G(x), or x-G(x)).**
>
> (i) A latent space preserving class-essential information is helpful to train a classifier **if it will not be attacked by any adversarial examples**. But this is not any of the problems we aim to solve in this paper, i.e., how to detect and defend the adversarial attacks. We study input-space disentanglement because the attacks perturb the input pixels.
>
> (ii) In addition, as revealed by most adversarial attack papers, such class-essential information is vulnerable to adversarial attacks and one should avoid using it for robust classification. Moreover, if the goal is to extract latent representations of task-essential information, one does not need to develop any special model: the information-bottleneck theory[1] suggests that training a DNN classifier automatically preserves the class-essential information only in its deeper layer representations.
>
> (iii) Furthermore, the classification does not have to be performed in the input space since our model can naturally produce low-dimensional class-disentangled representations, which are the bottleneck features $z$ of the VAE, since the class-redundant part $G(x)$ is solely generated from $z$. Hence, one can train a robust classifier taking $z$ instead of $G(x)$ as its input.
>
> **(4) The PGD attack is with respect to a pre-trained classifier on clean data x, not the classifier in CD-VAE on x-G(x), right? Because this will make sure x-G(x) is vulnerable to adversarial attack not merely because we are attacking D(x-G(x)).**
>
> Yes, the PGD attack is with respect to a classifier pre-trained on clean data $x$, which is irrelevant to the classifier used in CD-VAE. And they are separately (rather than jointly) trained. We will make this more clear.
>
> **(5) For adversarial attack restricted to lp-norm, there has been rich literatures on certified defense, i.e. defense with theoretical guarantee. Since this paper also focuses on lp-norm attack, formal theoretical discussions on the proposed method will greatly improve the paper. In line 116, the paper claims "we show that classification on the class-redundant part G(x) is robust to adversarial attacks". Is this true for all kinds of attack? What is the assumptions needed?**
>
> Our disentanglement is not generated for a specific attack. As evaluated on different types of attacks in Table 6-7 and analyzed by the empirical study in Sec. 3.2, the claim holds true for all the evaluated attacks, which **cover most types of attacks** adding small perturbations to the input of DNN classifiers. Sec. 3.2 presents thorough empirical analysis and verifies that the class-redundant part $G(x)$ contains useful information for classification but is very different from the information mainly distorted by adversarial attacks, i.e., $G(x)$ is robust to the attacks. In Table 7, our method successfully defends all the different adversarial attacks including $l _ p$-norm bounded attack, JPEG, ReColor and StADV while other defense methods are only effective on some of them. Our method achieves the highest "Unseen Attacks (mean)", which is the defense accuracy averaged over all the attacks that are not used for adversarial training of the defense model. This demonstrates the robustness of $G(x)$ to different attacks on real data. We will add more theoretical discussion in the future version.
>
> **(6)  And if we can extract class-redundant part in the latent space using IB, is that also robust to adversarial attack? Any experimental results on this?**
>
> According to information-bottleneck theory[1], the latent representations of a DNN may already **have most class-redundant information removed**, so it is difficult and unnecessary to extract class-redundant information using IB (again) in such latent space. To verify this, we conduct an experiment on CIFAR-10:
> We pre-train a classifier (WideResNet-28-10) on clean data $x$ and use it to extract latent features $f$. We then train a **CD-VAE on these latent features** and disentangle $f$ into a class-redundant part $G(f)$ and a class-essential part $R(f)$. Given an adversarial example $x'$, our defense feeds the class-redundant latent feature $G(f')$ to a classifier pre-trained on the clean data of $f$. Different from the defense on $G(x)$, We find this latent space defense fails towards all attacks including PGD, C\&W and StADV, achieving nearly 0.00\% defense accuracy. Meanwhile, it achieves high clean data accuracy of 95.19\%, indicating that $f$ only preserves class-relevant information. This result implies the necessity of input-space class-disentanglement for adversarial examples, as the original input contains both class-relevant and class-redundant information.
>
> **(7) In Eq 2, why do we need disentanglement (beta-VAE) in the feature space of VAE. Such a disentanglement notion seems irrelevant to the class disentanglement discussed in this paper.**
>
> Beta-VAE allows us to add randomness to $G(x)$ by sampling from the bottleneck feature distribution. The randomness is helpful to defend the deterministic adversarial attacks.
>
> **(8) No error bars are provided for all the tables, which may be needed to demonstrate that the empirical findings are not from randomness (especially for a pure empirical paper).**
>
> The only sources for randomness in our method are the model initialization and the random sampling of VAE's bottleneck embedding, which are common in VAE training. The table below reports the **error bars** (mean$\pm$std over five random trials) of AUC score for our detection method in Table 4:
>
> Method|FGSM|BIM|C\&W|PGD-$\ell _ \infty$|PGD-$\ell _ 2$
> ------|----|---|---|---|---
> KD ($R(x)$)|89.69$\pm$1.55|99.27$\pm$0.10|98.73$\pm$0.22|99.30$\pm$0.09|99.32$\pm$0.11|
> LID ($R(x)$)|98.59$\pm$0.59|97.29$\pm$0.94|95.10$\pm$0.91|97.58$\pm$0.98|97.38$\pm$0.90|
> MD ($R(x)$)|99.36$\pm$0.16|99.74$\pm$0.03|99.68$\pm$0.04 |99.79$\pm$0.03|99.77$\pm$0.04|
>
> It shows that the variance caused by the randomness is small, especially for MD ($R(x)$), which achieves <0.2\% standard deviation. Hence, our empirical findings are not from randomness.
>
> ***
>
> [1] Naftali Tishby, Noga Zaslavsky. Deep Learning and the Information Bottleneck Principle. IEEE ITW, 2015.

---

> > ### Comment · Reviewer_TtXL · 2021-08-23
> > **Response to Paper7396 Authors**
> >
> > Thanks for the thorough response. I feel most of my questions have been addressed, so I have increased my score.

---

> > > ### Author Response · Authors · 2021-08-26
> > > **Thanks a lot for your support!**
> > >
> > > We are glad to hear that most of your questions have been addressed in our response. Thank you very much for increasing your score and your support is very important to us. We greatly appreciate that!

---

### Decision · Program_Chairs · 2021-09-27

**Decision:**

Accept (Poster)

**Comment:**

The paper addresses adversarial detection and defense through a model that disentangles task-relevant and -irrelevant information. The model performs favorably against baselines, and the authors provided additional more recent baselines following feedback from reviewers. On the whole, the paper received a large number of reviews that were positive with engaging and informative discussion, so I believe that this paper has something positive to offer for the venue. There were some noted weaknesses to the paper; the reviewers believe the paper should still be accepted, but they ask that some of their requested changes make it to the final draft. I recommend acceptance as a poster.